# COKE: Core Kernel for More Efficient Approximation of Kernel Weights in Multiple Kernel Clustering

**Weixuan Liang** [1]  **Xinwang Liu** [1]  **Ke Liang** [1]  **Jiyuan Liu** [2]  **En Zhu** [1]

## Abstract

Inspired by the well-known coreset in clustering algorithms, we introduce the definition of the core kernel for multiple kernel clustering (MKC) algorithms. The core kernel refers to running MKC algorithms on smaller-scale base kernel matrices to obtain kernel weights similar to those obtained from the original full-scale kernel matrices. Specifically, the core kernel refers to a set of kernel matrices of size $\widetilde{\mathcal{O}}(1/\varepsilon^2)$ that perform MKC algorithms on them can achieve a $(1+\varepsilon)$-approximation for the kernel weights. Subsequently, we can leverage approximated kernel weights to obtain a theoretically guaranteed large-scale extension of MKC algorithms. In this paper, we propose a core kernel construction method based on singular value decomposition and prove that it satisfies the definition of the core kernel for three mainstream MKC algorithms. Finally, we conduct experiments on several benchmark datasets to verify the correctness of theoretical results and the efficiency of the proposed method.

## 1. Introduction

Multiple kernel clustering (MKC) algorithms (Zhao et al., 2009; Liu, 2023; 2022; Ren & Sun, 2020; Liu et al., 2017; 2016; Li et al., 2016; Feng et al., 2025), which have a strong capability to handle multi-source data, have been widely applied in various fields. MKC can be applied in various areas, including cancer biology (Gönen & Margolin, 2014), urban VANETs (Sellami & Alaya, 2021), healthcare (Che & Yang, 2024), network security (Hu et al., 2021), and others. However, due to the high computational complexity, MKC

[1]College of Computer Science and Technology, National University of Defense Technology, Changsha, China. [2]College of Systems Engineering, National University of Defense Technology, Changsha, China. Correspondence to: Xinwang Liu <xinwangliu@nudt.edu.cn>.

*Proceedings of the 42nd International Conference on Machine Learning*, Vancouver, Canada. PMLR 267, 2025. Copyright 2025 by the author(s).

struggles to handle large-scale datasets, making it difficult to meet the demands of the big data era.

To address the high complexity of MKC, this paper introduces the concept of core kernel, inspired by the idea of coreset (Har-Peled & Mazumdar, 2004; Chen, 2009). In the field of clustering, the coreset technique is an essential method for reducing the complexity of algorithms on large-scale datasets. A coreset is a weighted subset of the training set, such that the algorithm obtains a solution similar to the one derived from the entire training set. The coreset method has been proven to be effectively applicable to $k$-median (Sohler & Woodruff, 2018), $k$-means(Cohen-Addad et al., 2022), and kernel $k$-means clustering (Jiang et al., 2024).

In MKC, a fundamental assumption is that the optimal kernel matrix is a weighted combination of the base kernel matrices (Huang et al., 2012; Liu et al., 2016; Liu, 2022). Thus, kernel weights are a crucial parameter that can significantly impact the final clustering performance. Multiple kernel $k$-means (MKKM) (Huang et al., 2012) minimizes the objective with regard to the kernel weights and clustering partition. After the optimization of kernel weights, MKKM can obtain better clustering performance compared to using fixed weights. Subsequently, (Liu et al., 2016) introduces a matrix-induced regularization term for the kernel weights to increase the diversity of the consensus kernel matrix. To avoid kernel weights falling into poor local optima, (Liu, 2022) proposes a min-max optimization-based objective function, which enables learning better kernel weights. Therefore, for a good approximation method, it is necessary to better approximate the kernel weights of the original algorithm.

To quantitatively analyze the approximation degree of the kernel weights, we attempt to adapt the concept of "coreset" to MKC algorithms. We define smaller-scale base kernel matrices that can well approximate the kernel weights of the original base kernel matrices as the core kernel. The formal definition of the core kernel can be found in Definition 3.1. By using a core kernel of size $\widetilde{\mathcal{O}}(1/\varepsilon^2)$[1] as the input to MKC, we can obtain the $(1 + \varepsilon)$-approximation kernel weights. With the approximated kernel weights, we

---

[1]$\widetilde{\mathcal{O}}(\cdot)$ hides logarithmic terms.

design a large-scale extension method for MKC algorithms with a theoretical guarantee. Then, we propose an effective method for constructing the core kernel. Our method is inspired by the observation that the singular values of the column sampling matrix can effectively approximate the full kernel matrix. Specifically, we first select $s$ anchors, where $s$ is much less than the sample number $n$. For each base kernel, we construct a kernel similarity matrix whose elements are computed based on the full training set and the anchor set through the kernel function. We then use the right singular vectors and the singular values to construct an $s \times s$ core kernel, and this method is termed singular value decomposition-based core kernel (SVD-CK). The detailed process of the SVD-CK method is placed in Section 4.

In the existing literature on the large-scale extension of MKC, (Liang et al., 2023) achieves a good approximation of kernel weights by using random sampling. However, the method in (Liang et al., 2023) requires sampling many data points to achieve a sufficiently ideal approximation, which increases the computational cost during the algorithm's iterative process. As an improvement to the above method, (Liang et al., 2024) achieves better approximation results based on SVD, but its computational complexity is relatively high and related to $n$. The method proposed in this paper, SVD-CK, achieves an approximation performance comparable to that of (Liang et al., 2024). Moreover, during the algorithm's iterative process, it reduces the computational complexity to be independent of $n$. Additionally, the methods of (Liang et al., 2023; 2024) are solely designed to accelerate the SMKKM (Liu, 2022). In contrast, the method proposed in this paper can accelerate more MKC algorithms based on kernel weight learning, offering a broader range of applications.

Finally, experiments are conducted on several benchmark datasets to evaluate the approximation performance of SVD-CK on three mainstream MKC algorithms. The experimental results demonstrate that the proposed method can effectively approximate the kernel weights learned from the whole kernel matrix. Furthermore, the experiments also verify that the scalable extension method can be effectively applied to multiple large-scale datasets, proving the efficiency of the proposed approach.

## 2. Related Work

Before we introduce the related work, we briefly introduce basic assumptions and mathematical notations.

**Basic mathematical notations and assumptions.** We use $\| \cdot \|$ to present the spectral norm of a matrix or the 2-norm of a vector. For some vector $\mathbf{x}$, $\|\mathbf{x}\|_\infty = \max_i |x_i|$. $f(\cdot) \precsim g(\cdot)$ means $f(\cdot) \leq c g(\cdot)$ with some positive constant $c$. We provide definitions for all other symbols in their respective

contexts of use. For any kernel function used in this paper, we assume that $l \leq K(x, y) \leq b$ with positive constants $l, b$. The number of base kernels $m$ and clusters $k$ are both assumed to be constants.

### 2.1. Multiple Kernel Clustering

Multiple kernel clustering (MKC) (Huang et al., 2012) is an extension of kernel $k$-means (KKM) (Dhillon et al., 2004). Assume that the sample space is $\mathcal{X}$, the training set is $S = \{x_i\}_{i=1}^n \subseteq \mathcal{X}$, and the kernel function is $K : \mathcal{X} \times \mathcal{X} \to \mathbb{R}$. The objective function of KKM is

$$\min_{\mathbf{H}} \operatorname{tr}\left(\frac{1}{n}\mathbf{K}(\mathbf{I}_n - \mathbf{H}\mathbf{H}^\top)\right), \text{ s.t. } \mathbf{H}^\top\mathbf{H} = \mathbf{I}_k$$

where $\mathbf{H} \in \mathbb{R}^{n \times k}$ is termed clustering indicator matrix, and $\mathbf{K} \in \mathbb{R}^{n \times n}$ is the kernel matrix whose elements can be represented by $K_{ij} = K(x_i, x_j)$. One can perform eigen-decomposition on $\mathbf{K}$ and let $\mathbf{H}$ be the first $k$ largest eigenvectors. Then, the clustering results can be obtained by performing standard $k$-means on $\mathbf{H}$.

In the actual execution of KKM, we do not know which kernel function performs better. Therefore, we can select $m$ multiple kernel functions and compute base kernel matrices $\{\mathbf{K}_p\}_{p=1}^m$ accordingly. A fundamental assumption of the MKC algorithm is that the optimal kernel matrix is a weighted linear combination of the base kernel matrices. During the optimization process, the clustering indicator matrix and kernel weights are jointly optimized. In this section, we introduce two MKC algorithms. The first one is multiple kernel $k$-means (MKKM) (Huang et al., 2012). Denoting that $\Delta$ is the simplex constraint, the objective function of MKKM is

$$\begin{aligned} \min_{\boldsymbol{\gamma},\mathbf{H}} &\operatorname{tr}\left(\frac{1}{n}\mathbf{K}_{\boldsymbol{\gamma}}(\mathbf{I}_n - \mathbf{H}\mathbf{H}^\top)\right), \\ &\text{s.t. } \mathbf{H}^\top\mathbf{H} = \mathbf{I}_k, \boldsymbol{\gamma} \in \Delta, \end{aligned} \quad (1)$$

where $\mathbf{K}_{\boldsymbol{\gamma}} = \sum_{p=1}^m \gamma_p^2 \mathbf{K}_p$, and $\boldsymbol{\gamma} = [\gamma_1, \cdots, \gamma_m]^\top$ are the kernel weights. Another highly influential MKC algorithm is SMKKM (Liu, 2022), and its objective function is

$$\min_{\boldsymbol{\gamma}} f(\boldsymbol{\gamma}), \text{ s.t. } \boldsymbol{\gamma} \in \Delta, \quad (2)$$

where $f(\boldsymbol{\gamma}) = \max_{\mathbf{H}^\top\mathbf{H}=\mathbf{I}_k} \operatorname{tr}\left(\frac{1}{n}\mathbf{K}_{\boldsymbol{\gamma}}\mathbf{H}\mathbf{H}^\top\right)$. Whether MKKM or SMKKM, the computational complexity of obtaining the optimized kernel weights reaches $\mathcal{O}(n^3)$, which limits its application to large-scale datasets. It is noticed that MKC algorithms can also handle multi-view datasets, if we construct a kernel matrix for each view.

### 2.2. Coresets of Approximation Clustering

(Har-Peled & Mazumdar, 2004) introduces the concept of coreset for approximation clustering. A general definition of clustering is as follows.

**Definition 2.1** (Clustering Loss, (Har-Peled & Mazumdar, 2004)). For a set of points $S$ from sample space $\mathcal{X}$, with a weight function $w : S \rightarrow \mathbb{R}^+$ and clustering centroids $C$, let $v_C(S) = \sum_{x \in S} w(x)d(x, C)$ as the clustering loss of the $k$-median clustering caused by $C$, where $d(x, C) = \min_{y \in C} d(x, y)$ is the distance between $x$ and $C$. Similarly, denote that $\mu_C(S) = \sum_{x \in S} w(x)d(x, C)^2$ is the clustering loss of $k$-means clustering of $S$ caused by the clustering centroids $C$. Moreover, the clustering loss of the optimal $k$-median and $k$-means clustering for $S$ are respectively denoted by

$$
\begin{aligned}
v_{opt}(S, k) &= \min_{C \subseteq \mathcal{X}, |C|=k} v_C(S) \text{ and} \\
\mu_{opt}(S, k) &= \min_{C \subseteq \mathcal{X}, |C|=k} \mu_C(S).
\end{aligned}
\tag{3}
$$

The main idea of the coreset is to identify a small, weighted subset $T$ of the large dataset $S$, ensuring that performing a clustering task on this subset can yield an approximately optimal solution for the original dataset. The specific definition of coreset is as follows.

**Definition 2.2** (Coreset, (Har-Peled & Mazumdar, 2004)). A weighted set $T \subseteq \mathcal{X}$ is a $(k, \varepsilon)$-coreset of $S$ for the $k$-median clustering, if $\forall C \subseteq \mathcal{X}$ of $k$ points, the following equality holds,

$$
(1 - \varepsilon)v_C(S) \leq v_C(T) \leq (1 + \varepsilon)v_C(S).
$$

Similarly, $T$ is a $(k, \varepsilon)$-coreset of $S$ for the $k$-means clustering, if $\forall C \subseteq \mathcal{X}$, we have

$$
(1 - \varepsilon)\mu_C(S) \leq \mu_C(T) \leq (1 + \varepsilon)\mu_C(S).
$$

A coreset is a general data compression tool that allows clustering algorithms to run on smaller-scale datasets, enabling the attainment of a good approximate solution with reduced computational cost. In the MKC algorithm, the objective function typically lacks explicit clustering loss and cluster centroids, while the kernel weights play a critical role in determining the clustering performance. Therefore, this paper proposes the concept of "core kernel," inspired by the idea of the coreset, to approximate the kernel weights in the MKC algorithm.

# 3. Core Kernel and Its Application for Large-scale Extension

In this section, we introduce the core kernel definition and its application for the large-scale extension of MKC algorithms.

## 3.1. Definition of Core Kernel

**Definition 3.1.** Assume that $K_n = \{\frac{1}{n}\mathbf{K}_p\}_{p=1}^m \subseteq \mathbb{R}^{n \times n}$ is a set of base kernel matrices, and the kernel weights obtained

by performing some MKC algorithm on $\{\mathbf{K}_p\}_{p=1}^m$ are $\boldsymbol{\alpha}^*$. For some positive integer $s$, $\widetilde{K}_s = \{\widetilde{\mathbf{K}}_p\}_{p=1}^m \subseteq \mathbb{R}^{s \times s}$ is another set of kernel matrices, and the corresponding kernel weights are $\tilde{\boldsymbol{\alpha}}$ obtained from the same MKC algorithm. $\widetilde{K}_s$ is a $(1 + \varepsilon)$-approximation core kernel set of $K_n$, if $\|\tilde{\boldsymbol{\alpha}} - \boldsymbol{\alpha}\|_\infty \precsim \varepsilon$.

**Remark.** As seen, the core kernel is a concept proposed for the approximation of kernel weights. If the time complexity of some MKC algorithm is $\mathcal{O}(n^3)$, one can obtain the approximated kernel weights from the core kernel with time complexity $\mathcal{O}(s^3)$. When $s \ll n$, the time cost of the MKC algorithm can be dramatically reduced. Moreover, by incorporating the Nyström method (Wang et al., 2019), the construction of the core kernel enables the MKC algorithm to handle large-scale datasets, which we will introduce in the next subsection.

## 3.2. Large-scale Extension for MKC Algorithms

Now, we introduce how to use the core kernel set for the large-scale extension of MKC algorithms with Nyström method (Wang et al., 2019). Suppose that there is an anchor set (randomly sampled from the training set $S$) $\{a_1, \cdots, a_s\}$ and a core kernel set $\widetilde{K}_s = \{\widetilde{\mathbf{K}}_p\}_{p=1}^m$. For some MKC algorithms, we can use the core kernel set to obtain a group of the approximated kernel weights $\tilde{\boldsymbol{\alpha}}$. The complexity of MKC algorithms is usually $\mathcal{O}(s^3)$. Then, we construct $m$ kernel similarity matrices $\{\mathbf{P}_p\}_{p=1}^m \subseteq \mathbb{R}^{n \times s}$, where the element in the $i$-th row and $j$-th column of $\mathbf{P}_p$ is $K_p(x_i, a_j)$. Then, make a weighted combination of $\{\mathbf{P}_p\}_{p=1}^m$ by $\mathbf{P}_{\tilde{\boldsymbol{\alpha}}} = \sum_{p=1}^m \tilde{\alpha}_p^2 \mathbf{P}_p$. The summation of $\{\mathbf{P}_p\}_{p=1}^m$ costs $\mathcal{O}(nms)$ time. Then, we can perform SVD on $\mathbf{P}_{\tilde{\boldsymbol{\alpha}}}$, and obtain its first $k$ left singular vectors $\widetilde{\mathbf{H}}$. This step costs $\mathcal{O}(ns^2)$ time. Finally, we can obtain the clustering results by performing the standard $k$-means on $\widetilde{\mathbf{H}}$. Above all, the time cost is basically linear with the sample number $n$ (if $m, s \ll n$), and thus, it can be used to handle large-scale datasets. The above large-scale extension method is listed in Algorithm 1.

We will now conduct a theoretical analysis of the above algorithm. Before that, we need to introduce a common assumption.

**Assumption 3.2.** For any vector $\boldsymbol{\gamma} \in \mathbb{R}^m$, let the difference between the $j$-th and $(j + 1)$-th eigenvalues of the kernel matrix $\frac{1}{n}\mathbf{K}_{\boldsymbol{\gamma}}$ be denoted as $\delta_j(\boldsymbol{\gamma})$. For any $j \in [k]$ and any $\boldsymbol{\gamma} \in \Delta$, there exists a constant $c \geq 0$ such that $\delta_j(\boldsymbol{\gamma}) \geq 1/c$.

**Remark.** The assumption regarding eigenvalue gaps is quite common in matrix perturbation theory (Stewart, 1990). Specifically, when studying the perturbation of eigenvectors or orthogonal projections, researchers often assume that the gaps between eigenvalues are greater than a certain constant. (Von Luxburg et al., 2008) assumes that all eigenvalues of

**Algorithm 1** Large-Scale Extensions of MKC by Core Kernel

1: **Input:** Training set $m$ kernel functions $\{K_p(\cdot,\cdot)\}_{p=1}^m$; anchor sets $A = \{a_j\}_{j=1}^s$ (sampling from $S = \{x_i\}_{i=1}^n$ without replacement); core kernel set $\{\widetilde{\mathbf{K}}_p\}_{p=1}^m$; number of clusters $k$.

2: **Output:** clustering indicator matrix $\widetilde{\mathbf{H}}$; the clustering results.

3: Perform MKC algorithm on core kernel set to obtain approximated kernel weights $\tilde{\boldsymbol{\alpha}}$.

4: Compute $m$ base kernel similarity matrices $\{\mathbf{P}_p\}_{p=1}^m$ by $\mathbf{P}_p(i,j) = K_p(x_i, a_j)$, for any $i \in [n], j \in [s]$.

5: Make the weighted combination of $\{\mathbf{P}_p\}_{p=1}^m$ by $\mathbf{P}_{\tilde{\boldsymbol{\alpha}}} = \sum_{p=1}^m \tilde{\alpha}_p^2 \mathbf{P}_p$.

6: Perform SVD on $\mathbf{P}_{\tilde{\boldsymbol{\alpha}}}$ to obtain its first $k$ left singular vectors $\widetilde{\mathbf{H}} \in \mathbb{R}^{n \times k}$.

7: Perform $k$-means on $\widetilde{\mathbf{H}}$ for the final clustering results.

---

the Laplacian matrix are distinct, which is analogous to the assumption of eigenvalue gaps. In other research of kernel clustering (Liang et al., 2024; Mitz & Shkolnisky, 2022), the authors also make this assumption.

**Theorem 3.3.** *Under Assumption 3.2, denote that the kernel weights output by performing some MKC algorithm on the original base matrices $\{\mathbf{K}_p\}_{p=1}^m$ is $\boldsymbol{\alpha}$, and the corresponding clustering indicator matrix is $\mathbf{H}$, i.e., the first $k$ eigenvectors of $\mathbf{K}_{\boldsymbol{\alpha}}$. When the inputs of Algorithm 1 are a $(1+\varepsilon)$-approximation core kernel set, denote that the output kernel weights are $\tilde{\boldsymbol{\alpha}}$ which satisfies $\|\tilde{\boldsymbol{\alpha}} - \boldsymbol{\alpha}\|_\infty \precsim \varepsilon$, where $\precsim$ denotes inequality up to a constant factor. If the anchor number $s \geq c \log(n/\delta)/\varepsilon^2$, the clustering indicator matrix $\widetilde{\mathbf{H}}$ output by Algorithm 1 can make*

$$\left\| \widetilde{\mathbf{H}}\widetilde{\mathbf{H}}^\top - \mathbf{H}\mathbf{H}^\top \right\|_{\mathrm{F}} \precsim \varepsilon$$

*holds with probability at least $1 - \delta$.*

**Remark.** Theorem 3.3 gives an upper bound of the difference between the subspace spanned by $\widetilde{\mathbf{H}}$ and $\mathbf{H}$. When $\varepsilon$ is sufficiently small, the clustering performance by performing $k$-means on $\widetilde{\mathbf{H}}$ and $\mathbf{H}$ will be similar. Theorem 3.3 gives a theoretical guarantee that we can use the core kernel set and Algorithm 1 to approximate the original MKC algorithms effectively. The proof can be found in Section C.1 of the appendix.

## 4. Construction Method and Theoretical Analysis

In this section, we present a method for constructing the core kernel. We then provide a theoretical analysis and prove that our method can produce the core kernel set for several MKC algorithms.

### 4.1. Construction Idea

Now, we present the construction idea of the core kernel. Our main objective is to approximate the spectrum of a $n \times n$ kernel matrix by a $s \times s$ one.

Spectral approximation (Weinberger, 1974; Swartworth & Woodruff, 2023) of matrices is an essential field in linear algebra. It aims to approximate the eigenvalues or eigenvectors of large-scale matrices. Given a $n \times n$ kernel matrix $\frac{1}{n}\mathbf{K}$, computing the precise spectrum of $\frac{1}{n}\mathbf{K}$ is a massive problem when $n$ is large. Alternatively, we can use randomized methods to approximate the spectrum of $\frac{1}{n}\mathbf{K}$. The most straightforward and often effective method is uniform sampling. Specifically, let $\mathbf{T} \in \mathbb{R}^{n \times s}$ be a random sampling matrix, and every column of $\mathbf{T}$ has only one non-zero element. Assume that we uniformly sample $s$ indexes $\{i_1, \cdots, i_s\}$ from $\{1, \cdots, n\}$ without replacement. For the $j$-th column of $\mathbf{T}$, its elements can be represented as $T_{ij} = 1$, if $i = i_j$ and $T_{ij} = 0$, otherwise. Assuming that $\mathbf{W} = \mathbf{T}^\top \mathbf{K}\mathbf{T}$, then we can use the eigenvalues of $\frac{1}{s}\mathbf{W}$ to approximate the eigenvalues of $\frac{1}{n}\mathbf{K}$. Denoting that $\mathbf{P} = \mathbf{T}^\top \mathbf{K}$, another method is using the singular values of $\frac{1}{\sqrt{ns}}\mathbf{P}$ for the approximation of $\frac{1}{n}\mathbf{K}$'s eigenvalues.

**Empirical observations.** We conduct numerical experiments on two kernel datasets to verify the approximation effect of the above two methods. Flower17 and CCV are two commonly used multiple kernel datasets, and we aim to approximate the eigenvalues of their average kernel matrices. We then compute $\frac{1}{n}\mathbf{K}$'s largest $k$ eigenvalues $\{\lambda_j\}_{j=1}^k$ (in a descending order). For two approximation methods, we construct $\mathbf{T}$ randomly, and compute $\frac{1}{s}\mathbf{W}$'s largest $k$ eigenvalues along with $\frac{1}{\sqrt{ns}}\mathbf{P}$'s largest $k$ singular values. Fixed the anchor number $s$, we compute the difference between the precise eigenvalue and the approximated one for every $j \in [k]$. We let the maximal difference be the approximation error. We let $s$ vary in $\{50 : 50 : 1000\}$ and record the variations of the approximation errors. To reduce the randomness, we repeat the above experiments 30 times and plot the mean values in Figure 1. As seen from Figure 1, the approximation effect of $\frac{1}{\sqrt{ns}}\mathbf{P}$ is much better than $\frac{1}{s}\mathbf{W}$. However, the time consumed by SVD decomposition of $\frac{1}{\sqrt{ns}}\mathbf{P}$ is relatively high. Therefore, when constructing the core kernel, we aim to combine the strengths of two approximation methods, i.e., achieving an approximation of SVD decomposition using the matrix with size $s \times s$.

**Theoretical observations.** Next, we conduct a theoretical analysis of the approximation effect of $\frac{1}{\sqrt{ns}}\mathbf{P}$'s singular values on the eigenvalues of $\frac{1}{n}\mathbf{K}$. This is also crucial for our subsequent analysis of the properties of the core kernel. We have the following theorem.

**Theorem 4.1.** *Let $\mathbf{T} \in \mathbb{R}^{n \times s}$ be a random sampling matrix and for the $j$-th column ($j \in [s]$), $T_{ij} = 1$ with probability*

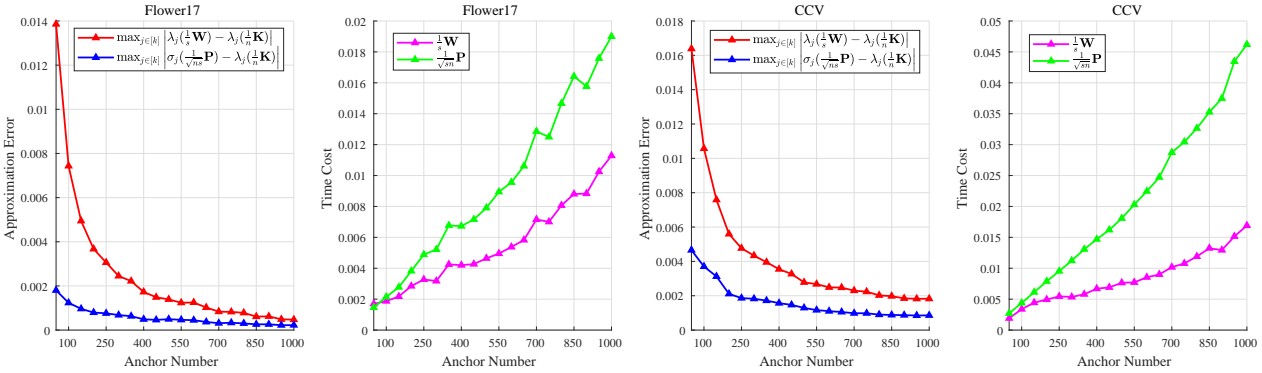

*Figure 1.* The comparison on the eigenvalue approximation errors of two methods, i.e., the eigenvalues of $\frac{1}{s}\mathbf{W}$ (red curves) and the singular values of $\frac{1}{\sqrt{ns}}\mathbf{P}$ (blue curves).

$1/n$ and $T_{ij} = 0$ *otherwise. Assume that* $\mathbf{P} = \mathbf{T}^\top \mathbf{K}$. *Then, if* $s \geq c \log(n/\delta)/\varepsilon^2$ *for some positive constant, with probability at least* $1 - \delta$, *for all* $t \in [n]$,

$$\left| \sigma_t \left( \frac{1}{\sqrt{ns}}\mathbf{P} \right) - \lambda_t \left( \frac{1}{n}\mathbf{K} \right) \right| \leq \varepsilon,$$

*where* $\sigma_t(\cdot)$ *is the $t$-th singular value of some matrix, and* $\lambda_t(\cdot)$ *is the $t$-th eigenvalue.*

**Remark.** Theorem 4.1 proves that using SVD decomposition can effectively approximate the eigenvalues of the whole kernel matrix, providing a theoretical foundation for constructing the core kernel. The proof of Theorem 4.1 can be found in Section C.2 of the appendix.

### 4.2. SVD-based Core Kernel

Based on the previous subsection, we begin constructing the core kernel and propose an algorithm based on SVD. For a better approximation, we use the singular values of $\frac{1}{\sqrt{ns}}\mathbf{P}$ to approximate the eigenvalues of $\frac{1}{n}\mathbf{K}$. Meanwhile, to reduce computational costs, we use the right singular vectors of $\frac{1}{\sqrt{ns}}\mathbf{P}$ as the eigenvectors of the core kernel.

Specifically, we first randomly select $s$ anchors $A = \{a_i\}_{i=1}^s$ from $S = \{x_i\}_{i=1}^n$. For the $p$-th base kernel, denoting that the corresponding kernel function is $K_p(\cdot, \cdot)$, we can compute $\mathbf{P}_p$ whose element can be computed by $\mathbf{P}_p(i, j) = K_p(x_i, a_j)$ $(i \in [n], j \in [s])$. Then, we perform eigen-decomposition on $\mathbf{P}_p^\top \mathbf{P}_p$ and denote $\mathbf{P}_p^\top \mathbf{P}_p = \mathbf{V}_p \mathbf{D}_p \mathbf{V}_p^\top$, where $\mathbf{V}_p \in \mathbb{R}^{s \times s}, \mathbf{D}_p \in \mathbb{R}^{s \times s}$. Notice that the diagonal elements of $\mathbf{D}_p^{1/2}$ is the first $s$ singular values of $\mathbf{P}_p$ and $\mathbf{V}_p$ is composed of the corresponding singular vectors. Then, we can construct the core kernel matrix of the $p$-th kernel by $\frac{1}{\sqrt{ns}}\widetilde{\mathbf{K}}_p = \frac{1}{\sqrt{ns}}\mathbf{V}_p \mathbf{D}_p^{1/2} \mathbf{V}_p^\top$. The pseudocode is provided in Algorithm 2. It can be seen that the algorithm we propose is very simple and easy to implement.

---

**Algorithm 2** SVD-based Core Kernel Construction

1: **Input:** Training set $S = \{x_i\}_{i=1}^n$, anchor set $A = \{a_i\}_{i=1}^s$, base kernel functions $\{K_p(\cdot, \cdot)\}_{p=1}^m$.
2: **Output:** Core kernel matrices $\{\widetilde{\mathbf{K}}_p\}_{p=1}^m$.
3: **for** $p = 1 : m$ **do**
4:      Compute $\mathbf{P}$ by $\mathbf{P}_p(i, j) = K_p(x_i, a_j)$.
5:      Perform eigen-decomposition on $\mathbf{P}_p^\top \mathbf{P}_p$ such that $\mathbf{P}_p^\top \mathbf{P}_p = \mathbf{V}_p \mathbf{D}_p \mathbf{V}_p^\top$.
6:      Let the $p$-th core kernel matrix be $\widetilde{\mathbf{K}}_p = \mathbf{V}_p \mathbf{D}_p^{1/2} \mathbf{V}_p^\top$.
7: **end for**

---

In this section, we first analyze the proposed Algorithm 2 from a theoretical perspective. Then, we utilize the core kernel for the large-scale extension of MKC algorithms and give the corresponding theoretical analysis.

### 4.3. Theoretical Analysis of SVD-based Core Kernel

The most significant difficulty of the analysis is the different sizes of the original kernel and the core kernel. To address this issue, we need to introduce the following empirical integral operator $L_K$ associated with $\frac{1}{n}\mathbf{K}$ (Von Luxburg et al., 2008).

$$\begin{aligned} L_K : \mathcal{C}(\mathcal{X}) &\to \mathcal{C}(\mathcal{X}), \\ L_K f(x) &= \frac{1}{n} \sum_{i=1}^n K(x, x_i) f(x_i), \end{aligned} \quad (4)$$

where $\mathcal{C}(\mathcal{X})$ is the space of continuous functions defined on $\mathcal{X}$. Then, $L_K$ and $\frac{1}{n}\mathbf{K}$ has the same non-zero eigenvalues and the eigenfunction of $L_K$ is

$$h_t(x) = \frac{1}{n\lambda_t} \sum_{i=1}^n K(x, x_i) h_t(x_i),$$

where $h_t(x_i) = \sqrt{n} h_{it}$, $h_{it}$ is the $i$-th component of $\mathbf{h}_t$, and $(\lambda_t, \mathbf{h}_t)$ is the $t$-th eigen-pair of $\frac{1}{n}\mathbf{K}$. A detailed introduction of the empirical integral operator and its perturbation property can be found in Section B.1 in the appendix.

We then rewrite the core kernel matrix into the form of a function, i.e., the kernel function associated with the core kernel $\frac{1}{\sqrt{ns}}\widetilde{\mathbf{K}}$. For some kernel function $K(\cdot, \cdot)$, assume that the corresponding feature map is $\phi(\cdot)$, i.e., $\phi^\top(x)\phi(y) = K(x, y)$. Denote that $\boldsymbol{\Phi}_n = [\phi(x_1), \cdots, \phi(x_n)]$ and $\boldsymbol{\Phi}_s = [\phi(a_1), \cdots, \phi(a_s)]$. It can be checked that

$$\frac{1}{\sqrt{ns}}\widetilde{\mathbf{K}} = \frac{1}{ns}\boldsymbol{\Phi}_s^\top \boldsymbol{\Phi}_n (\frac{1}{ns}\boldsymbol{\Phi}_n^\top \boldsymbol{\Phi}_s \boldsymbol{\Phi}_s^\top \boldsymbol{\Phi}_n)^{+1/2}\boldsymbol{\Phi}_n^\top \boldsymbol{\Phi}_s.$$

Denote $\boldsymbol{\Pi}' = \frac{1}{n}\boldsymbol{\Phi}_n(\frac{1}{ns}\boldsymbol{\Phi}_n^\top \boldsymbol{\Phi}_s \boldsymbol{\Phi}_s^\top \boldsymbol{\Phi}_n)^{+1/2}\boldsymbol{\Phi}_n^\top$, then the kernel function associated with $\widetilde{K}$ can be represented by

$$\widetilde{K}(x, y) = \phi^\top(x)\boldsymbol{\Pi}'\phi(y).$$

Assume that $(\tilde{\lambda}_t, \tilde{\mathbf{h}}_t)$ is the $t$-th eigen-pair of $\frac{1}{\sqrt{ns}}\widetilde{\mathbf{K}}$. We let the first $k$ eigenfunctions of $L_{\widetilde{K}}$ be $\{\tilde{h}_j(\cdot)\}_{j=1}^k$, i.e.,

$$\tilde{h}_j(x) = \frac{1}{s\tilde{\lambda}_t}\sum_{t=1}^s \widetilde{K}(x, a_t)\tilde{h}_j(a_t),$$

where $\tilde{h}_j(a_t) = \sqrt{s}\tilde{h}_{tj}$, and $\tilde{h}_{tj}$ is the $t$-th component of $\tilde{\mathbf{h}}_t$. Based on the above definitions of empirical operators and eigenfunctions, we can define the alignment level between the $p$-th base kernel function $K_p(\cdot, \cdot)$ and eigenfunctions $\{\hat{h}_j(\cdot)\}_{j=1}^k$ by

$$\mathcal{T}_n(K_p, \{\hat{h}_j\}_{j=1}^k) = \frac{1}{n^2}\sum_{j=1}^k \sum_{i,t=1}^n K_p(x_i, x_t)\hat{h}_j(x_i)\hat{h}_j(x_t).$$

Similarly, the alignment level between the $p$-th core kernel function $\widetilde{K}_p(\cdot, \cdot)$ and eigenfunctions $\{\tilde{h}_j(\cdot)\}_{j=1}^k$ can be given by

$$\mathcal{T}_s(\widetilde{K}_p, \{\tilde{h}_j\}_{j=1}^k) = \frac{1}{s^2}\sum_{j=1}^k \sum_{i,t=1}^s \widetilde{K}_p(a_i, a_t)\tilde{h}_j(a_i)\tilde{h}_j(a_t).$$

For any kernel weights $\boldsymbol{\gamma} = [\gamma_1, \cdots, \gamma_m]^\top$, letting $K_{\boldsymbol{\gamma}}(x, y) = \sum_{p=1}^m \gamma_p^2 K_p(x, y)$. Suppose that the corresponding eigenfunctions of the empirical integral operator $L_{K_{\boldsymbol{\gamma}}}$ are $\{\hat{h}_j^{\boldsymbol{\gamma}}\}_{j=1}^k$. Similarly, for the same kernel weights, suppose that the weighted combination of the core kernel matrices is $\widetilde{K}_{\boldsymbol{\gamma}}(x, y) = \sum_{p=1}^m \gamma_p^2 \widetilde{K}_p(x, y)$. We assume that the eigenfunctions of $L_{\widetilde{K}_{\boldsymbol{\gamma}}}$ are $\{\tilde{h}_j^{\boldsymbol{\gamma}}\}_{j=1}^k$. The following two lemmas give the upper bounds of the differences between the alignment level of the $p$-th base kernel and core base kernel with their corresponding eigenfunctions.

**Lemma 4.2.** *For any kernel weights $\boldsymbol{\gamma}$, when the number of anchors $s \geq c\log(n/\delta)/\varepsilon^2$ with some constant $c > 0$,*

$$|\mathcal{T}_n(K_p, \{\hat{h}_j^{\boldsymbol{\gamma}}\}_{j=1}^k) - \mathcal{T}_s(\widetilde{K}_p, \{\tilde{h}_j^{\boldsymbol{\gamma}}\}_{j=1}^k)| \leq k\varepsilon,$$

*holds with probability at least $1 - \delta$.*

By Lemma 4.2, we can derive the following Lemma 4.3 under Assumption 3.2.

**Lemma 4.3.** *Under Assumption 3.2, for any kernel weights $\boldsymbol{\alpha}, \boldsymbol{\beta}$, when the number of anchors $s \geq c\log(n/\delta)/\varepsilon^2$ with some constant $c > 0$,*

$$|\mathcal{T}_n(K_p, \{\hat{h}_j^{\boldsymbol{\alpha}}\}_{j=1}^k) - \mathcal{T}_s(\widetilde{K}_p, \{\tilde{h}_j^{\boldsymbol{\beta}}\}_{j=1}^k)| \leq \|\boldsymbol{\alpha} - \boldsymbol{\beta}\|_\infty + k\varepsilon,$$

*holds with probability at least $1 - \delta$.*

**Remark.** Lemma 4.2 gives the differences between the alignment level of the base kernel and core kernel for the same kernel weights. Furthermore, Lemma 4.3 gives the alignment differences with different weights. The proofs of the above lemmas are in C.3. By combining Lemma 4.3, we can utilize the recurrence relation to analyze the gradient differences of the base kernel and the core kernel during each step of the optimization process in the gradient descent-based MKC algorithms. Subsequently, we can prove that the SVD-based CK constructed by Algorithm 2 satisfies the definition of a core kernel as described in Definition 3.1 for SMKKM (Liu, 2022) and SMKKM-KWR (Li et al., 2023) (Theorem 4.4). Moreover, with some additional conditions, SVD-based CK is also the core kernel of MKKM-MR (Liu et al., 2016) (Theorem 4.5). The proofs of Theorem 4.4 and Theorem 4.5 are respectively placed in Section C.4 and Section C.5 of the appendix.

**Theorem 4.4.** *Under Assumption 3.2, if $s \geq c\log(n/\delta)/\varepsilon^2$, with probability at least $1 - \delta$, Algorithm 2 produces a $(1 + \varepsilon)$-approximation core kernel set for SMKKM and SMKKM-KWR.*

**Theorem 4.5.** *Denote that the elements of $\mathbf{M}, \widetilde{\mathbf{M}} \in \mathbb{R}^{m \times m}$ are respectively the Frobenius inner products of original and core base kernel matrices, i.e., $M_{pq} = \text{tr}(\frac{1}{n^2}\mathbf{K}_p \mathbf{K}_q)$ and $M_{pq} = \text{tr}(\frac{1}{ns}\widetilde{\mathbf{K}}_p \widetilde{\mathbf{K}}_q)$. Under Assumption 3.2, if $s \geq c\log(n/\delta)/\varepsilon^2$ and $\mathbf{M}, \widetilde{\mathbf{M}}$ have full ranks, with probability at least $1 - \delta$, Algorithm 2 produces a $(1 + \varepsilon)$-approximation core kernel set for MKKM-MK.*

## 5. Experiments

In this section, we conduct two kinds of experiments. The first one is to verify that Algorithm 2 can produce the core kernel set for SMKKM, SMKKM-KWR, and MKKM-MR. In the second kind of experiment, we then demonstrate that the core kernel set can also enable the above three MKC algorithms to handle large-scale datasets efficiently. All the above experiments are conducted on a computer with a configuration of Intel(R) Core(TM)-i7-10870H CPU.

## 5.1. Information of the Kernel Datasets

To verify the approximation effect of the core kernel on the kernel weights, we selected six small-scale kernel datasets for experimentation, including *Flower17*, *Digit*, *CCV*, *Flower102*, *4Area*, and *Cal102*. Their links and detailed information are reported in Section D.2 of the appendix.

## 5.2. Approximation Effect of Core Kernel on Kernel Weights

**Experimental setting.** We conduct experiments on three MKC algorithms, i.e., SMKKM, SMKKM-KWR and MKKM-MR. For the methods with hyper-parameters, we let all of the hyper-parameters be equal to $1$. We first perform the MKC algorithms on the original kernel datasets to obtain a set of kernel weights, denoted as $\boldsymbol{\alpha}$. Then, we randomly selected $s$ distinct numbers $\{i_1, \cdots, i_s\}$ from $\{1, \cdots, n\}$, where $n$ is the number of samples in the training set. Then, we use the indices $\{i_1, \cdots, i_s\}$ to construct a core kernel set by Algorithm 2. We perform the MKC algorithm on the core kernel set and denote the corresponding kernel weights by $\tilde{\boldsymbol{\alpha}}$. Let The value of $s$ vary within the range $[50 : 50 : 1000]$, with the constraint that $s$ is less than the number of samples but greater than the number of clusters. For each $s$, we record the value of $\|\tilde{\boldsymbol{\alpha}} - \boldsymbol{\alpha}\|_\infty$. To reduce randomness, we repeated the experiment 30 times and computed the average of $\|\tilde{\boldsymbol{\alpha}} - \boldsymbol{\alpha}\|_\infty$. The experimental results are shown as the blue curve in Figure 2. Additionally, for comparison, we construct $s \times s$ base kernel matrices via uniform sampling from the selected indices $\{i_1, \cdots, i_s\}$, selecting the corresponding rows and columns. We also recorded the difference between the kernel weights obtained from the original kernel matrices and the kernel matrices based on uniform sampling. After repeating the experiment 30 times, the average value is shown as the red curve in Figure 2.

**Experimental results.** Due to space limitations, only the experimental results on two datasets, i.e., Flower17 and DIGIT, are presented in the main text, while the results on other datasets can be found in Section D.1 of the appendix. From the blue curve, it can be observed that as $s$ increases, the kernel weights obtained by the algorithm on the proposed SVD-CK rapidly approach those obtained on the original kernel matrices. This fully demonstrates the correctness of Theorem 4.4 and Theorem 4.5. The red curve represents the kernel weight error obtained by the algorithm on the kernel matrices based on uniform sampling. It can be seen that the proposed method significantly outperforms uniform sampling. In addition, a relatively small $s$ can achieve a low approximation error on kernel weights, which highlights the effectiveness of SVD-CK in enabling scalable extensions of MKC algorithms.

## 5.3. Large-Scale Experiments

*Table 1.* Large-scale datasets

| Dataset | Samples | Number of | | Features |
| | | Views | Clusters | |
|---|---|---|---|---|
| CIFAR10 | 50000 | 3 | 10 | 512,2048,1024 |
| MNIST | 60000 | 3 | 10 | 342, 1024, 64 |
| Winnipeg | 325834 | 2 | 7 | 49, 38 |

**Experimental setting.** To validate the effectiveness of Algorithm 1, this section also conducts tests on several commonly used large-scale datasets, including *CIFAR10*[2], *MNIST*[3], and *Winnipeg*[4]. Their detailed information is reported in Table 1. The number of samples in the datasets used in the experiments exceeds $50,000$, with the largest being $325,834$. For each view, a base kernel similarity matrix is constructed using a Gaussian kernel function as follows:

$$K(x_i, a_t) = \exp\left(-\frac{\|x_i - a_t\|^2}{2\sigma^2}\right), \qquad (5)$$

where $x_i \in S(i \in [n])$ and $a_t \in A(t \in [s])$. In the proposed algorithm, $s$ is set to $s = 500$. The parameter $\sigma^2$ represents the average squared distance between the sample points in $S$ and $A$, , and is computed as:

$$\sigma^2 = \frac{1}{ns}\sum_{x_i \in S}\sum_{a_t \in A}\|x_i - a_t\|^2. \qquad (6)$$

*Table 2.* Results of large-scale experiments

| Datasets | CIFAR10 | MNIST | Winnipeg |
|---|---|---|---|
| | NMI (%) | | |
| RMKMC | 82.07 | 81.05 | 49.43 |
| LMVSC | 45.04 | 84.75 | 51.94 |
| OPMC | 83.81 | 82.67 | 50.82 |
| AWMVC | 76.38 | 80.76 | 38.86 |
| SMKKM (CK) | 97.53 | 97.00 | 54.14 |
| SMKKM-KWR (CK) | 97.78 | 96.96 | 54.12 |
| MKKM-MR (CK) | **98.07** | **97.33** | **59.24** |
| | Time (s) | | |
| RMKMC | 162.09 | 155.16 | 297.40 |
| LMVSC | 16.22 | 67.44 | 142.63 |
| OPMC | 27.56 | 49.94 | 20.29 |
| AWMVC | 203.01 | 64.78 | 59.77 |
| SMKKM (CK) | 47.84 | 65.18 | 288.06 |
| SMKKM-KWR (CK) | 43.61 | 65.77 | 248.51 |
| MKKM-MR (CK) | 38.99 | 62.26 | 259.24 |

For comparison, experiments are also conducted on several state-of-the-art large-scale multi-view clustering algorithms,

---

[2] http://www.cs.toronto.edu/~kriz/cifar.html
[3] http://yann.lecun.com/exdb/mnist/
[4] https://archive.ics.uci.edu/dataset/525/crop+mapping+using+fused+optical+radar+data+set

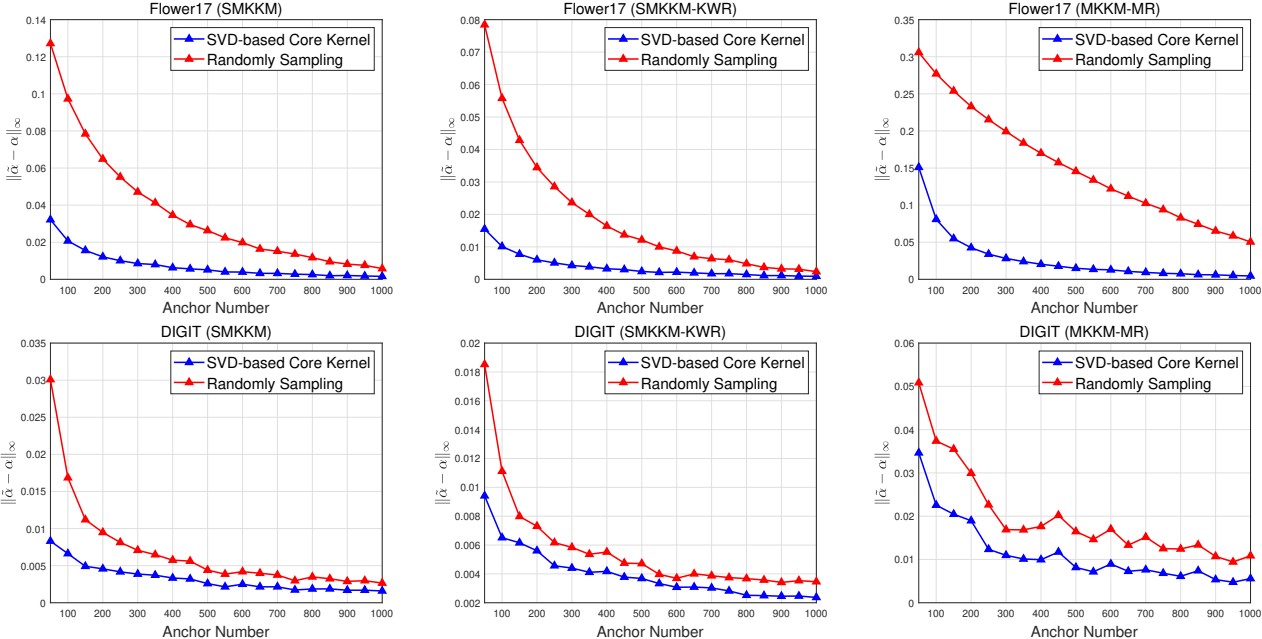

*Figure 2.* The proposed SVD-CK is illustrated through a diagram showing the kernel weight approximation performance. The blue curve represents the kernel weight approximation error constructed using SVD-CK. It can be observed that as $s$ increases, the approximation error decreases rapidly, enabling the weights obtained by the three MKC methods on SVD-CK to closely approximate those on the original kernel matrices. For comparison, the red curve represents the kernel weight approximation error based on random sampling of the kernel matrix. SVD-CK demonstrates a clear advantage in kernel weight approximation.

including: RMKMC (Cai et al., 2013), LMVSC (Kang et al., 2020), OPMC (Liu et al., 2021), and AWMVC (Wan et al., 2024). Detailed information on these comparison methods is reported in Section D.3 of the appendix. For the above comparison algorithms with hyper-parameters, the optimal hyper-parameters are selected via grid search as described in the corresponding papers. Our experiments employ three widely used clustering metrics: accuracy (ACC), normalized mutual information (NMI), and purity. Additionally, we record the execution time for all experiments. Due to limited space, we only show NMI and execution time in the main text. We use Algorithm 1 for the large-scale extensions of SMKKM, SMKKM-KWR, and MKKM-MR, and they are termed SMKKM (CK), SMKKM-KWR (CK), and MKKM-MR (CK), respectively. The experimental results are presented in Table 4, with the best outcomes highlighted in bold. For the whole experimental results, please refer to Section D.4 of the appendix.

As shown in Table 4, the proposed method enables the three MKC algorithms to operate on large-scale datasets. From the perspective of clustering performance, the three MKC methods demonstrate better clustering results compared to several large-scale multi-view clustering algorithms that directly process the original features of the data. This is because the kernel functions are effectively utilized, allowing better handling of non-linearly separable datasets. From

the perspective of clustering efficiency, the proposed large-scale extension of the MKC algorithms can obtain clustering results quickly, indicating that the computational cost is relatively low. The above experimental results fully demonstrate the effectiveness and efficiency of Algorithm 1.

## 6. Conclusion

This paper introduces a new concept, the core kernel, to address kernel weight approximation in multiple kernel clustering algorithms. We define the core kernel and, based on this definition, propose a theoretically guaranteed large-scale extension method for MKC. Subsequently, we introduce SVD-CK, a core kernel construction method based on singular value decomposition. We prove that SVD-CK satisfies the definition of the core kernel for the three MKC algorithms. Finally, we validate the approximation performance of SVD-CK for kernel weights on several commonly used kernel datasets. Additionally, on large-scale datasets, we verify the effectiveness and efficiency of the proposed large-scale extension method. Although this paper only explores the approximation of MKC, the proposed method demonstrates strong potential for broader applications. In particular, it could be extended to analyze the approximation algorithms of multi-view clustering (Yu et al., 2024; 2023), which is a direction we intend to explore in future work.

## Acknowledgments

This work is supported by the National Science Fund for Distinguished Young Scholars of China (No. 62325604), the National Natural Science Foundation of China (No. 62306324, U24A20333, 62441618, and 62276271), and the Science and Technology Innovation Program of Hunan Province (No. 2024RC3128).

## Impact Statement

This paper presents work whose goal is to advance the field of Machine Learning. There are many potential societal consequences of our work, none of which we feel must be specifically highlighted here.

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

# A. Brief Introduction of SMKKM, SMKKM-KWR, and MKKM-MR

In this section, we introduce SMKKM (Liu, 2022), SMKKM-KWR (Li et al., 2023), and MKKM-MR (Liu et al., 2016) and their optimization method.

**1. SMKKM.** (Liu, 2022) The objective function of SMKKM is

$$\min_{\boldsymbol{\gamma}} f(\boldsymbol{\gamma}), \text{ s.t. } \boldsymbol{\gamma} \in \Delta, \tag{7}$$

where $f(\boldsymbol{\gamma}) = \max_{\mathbf{H}^\top \mathbf{H} = \mathbf{I}_k} \operatorname{tr}\left(\frac{1}{n} \mathbf{K}_{\boldsymbol{\gamma}} \mathbf{H}\mathbf{H}^\top\right)$. The optimization of SMKKM is based on a reduced gradient descent method. Specifically, fixed some index $u \in [m]$, the reduced gradient of $f_{\boldsymbol{\gamma}}$ is as follows,

$$[\nabla f]_p = \frac{\partial f(\boldsymbol{\gamma})}{\partial \gamma_p} - \frac{\partial f(\boldsymbol{\gamma})}{\partial \gamma_u}, \forall p \neq u, \quad [\nabla f]_u = \sum_{p \neq u} \left(\frac{\partial f(\boldsymbol{\gamma})}{\partial \gamma_u} - \frac{\partial f(\boldsymbol{\gamma})}{\partial \gamma_p}\right), \tag{8}$$

where $\frac{\partial f(\boldsymbol{\gamma})}{\partial \gamma_p} = \frac{2\gamma_p}{n} \operatorname{tr}(\mathbf{K}_p(\mathbf{I}_n - \hat{\mathbf{H}}\hat{\mathbf{H}}^\top))$, and $\hat{\mathbf{H}} = \arg\min_{\mathbf{H}^\top \mathbf{H} = \mathbf{I}_k} \operatorname{tr}(\mathbf{K}_{\boldsymbol{\gamma}}(\mathbf{I}_n - \mathbf{H}\mathbf{H}^\top))$.

To keep the positivity constraint of $\boldsymbol{\gamma}$, the descent direction $\mathbf{d} = [d_1, \cdots, d_m]^\top$ can be set as

$$d_p = \begin{cases} 0, & \text{if } \gamma_p = 0 \text{ and } \frac{\partial f(\boldsymbol{\gamma})}{\partial \gamma_p} - \frac{\partial f(\boldsymbol{\gamma})}{\partial \gamma_u} > 0, \\ -\frac{1}{m-1}\left(\frac{\partial f(\boldsymbol{\gamma})}{\partial \gamma_p} - \frac{\partial f(\boldsymbol{\gamma})}{\partial \gamma_u}\right), & \text{if } \gamma_p > 0 \text{ and } p \neq u, \\ -\frac{1}{m-1} \sum_{p \neq u, \gamma_p > 0}\left(\frac{\partial f(\boldsymbol{\gamma})}{\partial \gamma_u} - \frac{\partial f(\boldsymbol{\gamma})}{\partial \gamma_p}\right), & \text{for } p = u. \end{cases} \tag{9}$$

Compared with (Liu, 2022), in this paper, the reduced gradient is divided by $m-1$ for the normalization of the $u$-th component. Nevertheless, the reduced gradient of this paper still makes $f(\boldsymbol{\gamma})$ converge within several iterations. The updating scheme is $\boldsymbol{\gamma} = \boldsymbol{\gamma} + \eta \mathbf{d}$, where $\eta$ is assumed to be less than some constant $c > 0$.

**2. SMKKM-KWR.** (Li et al., 2023) SMKKM-KWR is an improvement of SMKKM, and its objective function is

$$\min_{\boldsymbol{\gamma}} f(\boldsymbol{\gamma}), \text{ s.t. } \boldsymbol{\gamma} \in \Delta, \tag{10}$$

where $f(\boldsymbol{\gamma}) = \max_{\mathbf{H}^\top \mathbf{H} = \mathbf{I}_k} \operatorname{tr}\left(\frac{1}{n} \mathbf{K}_{\boldsymbol{\gamma}} \mathbf{H}\mathbf{H}^\top\right) + \lambda \|\boldsymbol{\gamma} - \boldsymbol{\gamma}_0\|^2$, where $\boldsymbol{\gamma}_0$ denotes the average kernel weights. The $p$-th component of the gradient is $\frac{\partial f(\boldsymbol{\gamma})}{\partial \gamma_p} = \frac{2\gamma_p}{n} \operatorname{tr}(\mathbf{K}_p(\mathbf{I}_n - \hat{\mathbf{H}}\hat{\mathbf{H}}^\top)) + 2\lambda(\gamma_p - \gamma_{0p})$, and $\hat{\mathbf{H}} = \arg\min_{\mathbf{H}^\top \mathbf{H} = \mathbf{I}_k} \operatorname{tr}(\mathbf{K}_{\boldsymbol{\gamma}}(\mathbf{I}_n - \mathbf{H}\mathbf{H}^\top))$. Similar to SMKKM, SMKKM-KWR can be optimized using the reduced gradient descent algorithm.

**3. MKKM-MR.** (Liu et al., 2016) MKKM-MR is an enhanced version of MKKM, and the objective function is

$$\min_{\boldsymbol{\gamma}, \mathbf{H}} \operatorname{tr}\left(\frac{1}{n} \mathbf{K}_{\boldsymbol{\gamma}}(\mathbf{I}_n - \mathbf{H}\mathbf{H}^\top)\right) + \lambda \boldsymbol{\gamma}^\top \mathbf{M} \boldsymbol{\gamma}, \text{ s.t. } \boldsymbol{\gamma} \in \Delta, \mathbf{H}^\top \mathbf{H} = \mathbf{I}_k,$$

where $\lambda$ is a hyper-parameter, $\mathbf{M} \in \mathbb{R}^{m \times m}$, and its element can be represented by $M_{pq} = \operatorname{tr}\left(\frac{1}{n^2} \mathbf{K}_p \mathbf{K}_q\right)$ (for $p, q \in [m]$). The optimization of MKKM-MR is based on a coordinate descent method as follows.

**1) Optimize H with fixed $\boldsymbol{\gamma}$.** Perform the eigen decomposition on $\mathbf{K}_{\boldsymbol{\gamma}}$, and let $\mathbf{H}$ be its first $k$ eigenvectors.

**2) Optimize $\boldsymbol{\gamma}$ with fixed H.** Let $\delta_p = \operatorname{tr}\left(\frac{1}{n} \mathbf{K}_p(\mathbf{I}_n - \mathbf{H}\mathbf{H}^\top)\right)$ and $\mathbf{D} \in \mathbb{R}^{m \times m}$ be a diagonal matrix with $D_{pp} = \delta_p$ (for $p \in [m]$). Then, $\boldsymbol{\gamma} = \frac{(\lambda\mathbf{M} + \mathbf{D})^{-1} \mathbf{1}_m}{\mathbf{1}_m^\top (\lambda\mathbf{M} + \mathbf{D})^{-1} \mathbf{1}_m}$ is the optimal solution.

## B. Preliminaries of Proofs

### B.1. Empirical Integral Operator and Perturbation Theory

We first introduce the empirical integral operator (Von Luxburg et al., 2008) associated with some kernel matrix $\frac{1}{n}\mathbf{K} \in \mathbb{R}^{n \times n}$. $\frac{1}{n}\mathbf{K} \in \mathbb{R}^{n \times n}$ can be regarded as an operator from $\mathbb{R}^n$ to $\mathbb{R}^n$, i.e.,

$$\frac{1}{n}\mathbf{K}\mathbf{w} = \left[\frac{1}{n}\sum_{i=1}^{n} K(x_1, x_i)w_i, \cdots, \frac{1}{n}\sum_{i=1}^{n} K(x_n, x_i)w_i\right]^{\top},$$

for any $\mathbf{w} = [w_1, \cdots, w_n]^{\top} \in \mathbb{R}^n$. Then, the empirical integral operator $L_K$ associated with $\frac{1}{n}\mathbf{K}$ is

$$L_K : \mathcal{C}(\mathcal{X}) \to \mathcal{C}(\mathcal{X}), \ L_K f(x) = \frac{1}{n}\sum_{i=1}^{n} K(x, x_i)f(x_i),$$

where $\mathcal{C}(\mathcal{X})$ denotes the space of continuous functions defined on $\mathcal{X}$. $L_K$ has the same non-zero eigenvalues with $\frac{1}{n}\mathbf{K}$. Let $\{\lambda_1, \cdots, \lambda_l\}$ (in a descending order) be the non-zero eigenvalues of $L_K$ and $\frac{1}{n}\mathbf{K}$. Assume that the corresponding eigenvectors of $\frac{1}{n}\mathbf{K}$ are $\{\mathbf{h}_1, \cdots, \mathbf{h}_l\}$. Then, the $t$-th ($t \in [l]$) eigenfunction $h_t$ of $L_K$ is

$$h_t(x) = \frac{1}{n\lambda_t}\sum_{i=1}^{n} K(x, x_i)h_t(x_i),$$

where $h_t(x_i) = \sqrt{n}h_{it}$, and $h_{it}$ is the $i$-th component of $\mathbf{h}_t$. The following theorem gives a perturbation bound of the empirical integral operator.

**Lemma B.1** (Theorem 7, (Von Luxburg et al., 2008)). *Let $(E, \|\cdot\|_E)$ be a Banach space, and let $B$ denote the unit ball in this space. Let $(K_n)_{n \in \mathbb{N}^+}$ and $K$ be compact operators on $E$, with $K_n$ converging to $K$. For a non-zero eigenvalue $\lambda \in \sigma(K)$, let $Pr$ denote its corresponding spectral projection. Let $M \subset \mathbb{C}$ be an open neighborhood of $\lambda$ such that $\sigma(K) \cap M = \lambda$. There exists an integer $N \in \mathbb{N}$ such that for $\forall n > N$, $\sigma(K_n) \cap M = \lambda$ is isolated in $\sigma(K_n)$. Let $Pr_n$ denote the spectral projection corresponding to $\sigma(K_n) \cap M$ for $K_n$. Then there exists a constant $C > 0$ such that for every $x \in PrE$, the following holds:*

$$\|x - Pr_n x\|_E \leqslant C(\|(K_n - K)x\|_E + \|x\|_E \|(K - K_n)K_n\|). \tag{11}$$

### B.2. Concentration Inequalities for Matrices and Vectors

In our proofs, we need the following three concentration inequalities: The first one gives a matrix Chernoff bound for the eigenvalues of sums of finite random matrices. The second and third are two inequalities of subsampled covariance matrices and vectors, respectively.

**Theorem B.2** (Chernoff bound of eigenvalues (Bakshi et al., 2020)). *Assume that $\{\mathbf{A}_j\}_{j \geq 1}$ is a finite sequence of independent, random, positive-semidefinite matrices with size $n \times n$. If $\|\mathbf{A}_j\| \leq L$ ($\forall j$) for some positive real number $L$ almost surely, then the following tail inequalities hold*

$$\begin{cases} \Pr\left[\lambda_k(\sum_j \mathbf{A}_j) \geq (1+\delta)\mu_k\right] \leq (n-k+1) \cdot \left[\frac{e^\delta}{(1+\delta)^{1+\delta}}\right]^{\mu_k/L}, & \text{for } \delta > 0, \\ \Pr\left[\lambda_k(\sum_j \mathbf{A}_j) \leq (1-\delta)\mu_k\right] \leq k \cdot \left[\frac{e^{-\delta}}{(1-\delta)^{1-\delta}}\right]^{\mu_k/L}, & \text{for } \delta \in [0, 1), \end{cases} \tag{12}$$

*where $\mu_k = \lambda_k(\sum_j \mathbb{E}[\mathbf{A}_j])$, and $k(\leq n)$ is some integer.*

**Theorem B.3** (Lemma2, (Bach, 2013)). *Let $\mathbf{\Psi}_n = [\boldsymbol{\psi}_1, \cdots, \boldsymbol{\psi}_n] \in \mathbb{R}^{r \times n}$, and $\|\boldsymbol{\psi}_i\| \leq R$, for each $i \in [n]$. Let $I$ be an index set that consists of $s$ elements sampled from $\{1, \cdots, n\}$ without replacement. Then, for all $\varepsilon > 0$,*

$$\Pr\left[\left\|\frac{1}{n}\mathbf{\Psi}_n\mathbf{\Psi}_n^{\top} - \frac{1}{s}\mathbf{\Psi}_I\mathbf{\Psi}_I^{\top}\right\| > \varepsilon\right] \leq r\exp\left(\frac{-s\varepsilon^2/2}{\left\|\frac{1}{n}\mathbf{\Psi}_n\mathbf{\Psi}_n^{\top}\right\| \cdot (R^2 + t/3)}\right).$$

**Theorem B.4** (Lemma1, (Smale & Zhou, 2007)). *Let $\mathcal{H}$ be a Hilbert space and $\{\boldsymbol{\psi}_i\}_{i=1}^{s}$ be $s$ i.i.d. random variables valued in $\mathcal{H}$. Assume that $\|\boldsymbol{\psi}_i\| \leq R$ with some constant $R > 0$. Denote that $\sigma^2 = \mathbb{E}(\|\boldsymbol{\psi}_i\|^2)$. Then,*

$$\Pr\left[\left\|\frac{1}{s}\sum_{i=1}^{s}(\boldsymbol{\psi}_i - \mathbb{E}[\boldsymbol{\psi}_i])\right\| \geq \varepsilon\right] \leq 2\exp\left(-\frac{s\varepsilon^2}{2R\varepsilon + 2\sigma^2}\right).$$

## C. Proofs of Theoretical Results

### C.1. Proof of Theorem 3.3

To prove Theorem 3.3, we need the following lemma on the upper bound of matrix eigenvector perturbation.

**Lemma C.1.** *(Yu et al., 2014) Let $\mathbf{A}, \mathbf{B} \in \mathbb{R}^{n \times n}$ be Hermitian matrices with eigenvalues $\lambda_1 \geq \cdots \geq \lambda_n$ and $\hat{\lambda}_1 \geq \cdots \geq \hat{\lambda}_n$, respectively. Fix $1 \leq r \leq s \leq n$, and assume $\min(\lambda_{r-1} - \lambda_r, \lambda_s - \lambda_{s+1}) > 0$, where $\lambda_0 := \infty$ and $\lambda_{n+1} := -\infty$. Let $d := s - r + 1$. Assume that $\mathbf{H} = [\mathbf{h}_r, \mathbf{h}_{r+1}, \cdots, \mathbf{h}_s] \in \mathbb{R}^{n \times d}$ and $\hat{\mathbf{H}} = [\hat{\mathbf{h}}_r, \hat{\mathbf{h}}_{r+1}, \cdots, \hat{\mathbf{h}}_s] \in \mathbb{R}^{n \times d}$ are column-orthogonal and satisfy, for any $j \in \{r, r+1, \cdots, s\}$, $\mathbf{A}\mathbf{h}_j = \lambda_j \mathbf{h}_j$ and $\mathbf{B}\hat{\mathbf{h}}_j = \hat{\lambda}_j \hat{\mathbf{h}}_j$. Then, there exists an orthogonal matrix $\hat{\mathbf{O}} \in \mathbb{R}^{d \times d}$ such that*

$$\left\| \hat{\mathbf{H}}\hat{\mathbf{O}} - \mathbf{H} \right\|_{\mathrm{F}} \leqslant \frac{2^{3/2} \min(d^{1/2} \|\mathbf{A} - \mathbf{B}\|, \|\mathbf{A} - \mathbf{B}\|_{\mathrm{F}})}{\min(\lambda_{r-1} - \lambda_r, \lambda_s - \lambda_{s+1})}. \tag{13}$$

*Proof.* Let $\overline{\mathbf{H}}$ be the first $k$ eigenvectors of $\frac{1}{n}\mathbf{K}_{\tilde{\boldsymbol{\alpha}}}$. For any orthogonal matrix $\hat{\mathbf{O}} \in \mathbb{R}^{k \times k}$, we have

$$\begin{aligned} & \left\| \overline{\mathbf{H}}\,\overline{\mathbf{H}}^\top - \mathbf{H}\mathbf{H}^\top \right\|_{\mathrm{F}} \\ \leq\, & \left\| \overline{\mathbf{H}}\hat{\mathbf{O}}\hat{\mathbf{O}}^\top \overline{\mathbf{H}}^\top - \overline{\mathbf{H}}\hat{\mathbf{O}}\mathbf{H}^\top \right\|_{\mathrm{F}} + \left\| \overline{\mathbf{H}}\hat{\mathbf{O}}\mathbf{H} - \mathbf{H}\mathbf{H}^\top \right\|_{\mathrm{F}} \\ \leq\, & \|\overline{\mathbf{H}}\hat{\mathbf{O}}\| \cdot \left\| \overline{\mathbf{H}}\hat{\mathbf{O}} - \mathbf{H} \right\|_{\mathrm{F}} + \|\mathbf{H}\| \cdot \left\| \overline{\mathbf{H}}\hat{\mathbf{O}} - \mathbf{H} \right\|_{\mathrm{F}} \\ \leq\, & 2 \left\| \overline{\mathbf{H}}\hat{\mathbf{O}} - \mathbf{H} \right\|_{\mathrm{F}}. \end{aligned} \tag{14}$$

By setting $r = 1, s = k$ in Lemma C.1, according to Assumption 3.2,

$$\begin{aligned} \left\| \overline{\mathbf{H}}\,\overline{\mathbf{H}}^\top - \mathbf{H}\mathbf{H}^\top \right\|_{\mathrm{F}} &\precsim \left\| \overline{\mathbf{H}}\hat{\mathbf{O}} - \mathbf{H} \right\|_{\mathrm{F}} \\ &\leq \frac{\left\| \frac{1}{n}\mathbf{K}_{\tilde{\boldsymbol{\alpha}}} - \frac{1}{n}\mathbf{K}_{\boldsymbol{\alpha}} \right\|_{\mathrm{F}}}{\delta(\boldsymbol{\alpha})} \\ &\precsim \sqrt{\sum_{p=1}^{m} \sum_{i=1}^{n} \sum_{t=1}^{n} (\tilde{\alpha}_p^2 - \alpha_p^2)^2 \frac{K_p^2(x_i, x_t)}{n^2}} \\ &\precsim \sqrt{\sum_{p=1}^{m} (\tilde{\alpha}_p^2 - \alpha_p^2)^2 \cdot \left( \max_{i \in [n], t \in [n]} K^2(x_i, x_t) \right)} \\ &\precsim \max_{p \in [m]} |\tilde{\alpha}_p - \alpha_p| \sqrt{\sum_{p=1}^{m} (\tilde{\alpha}_p + \alpha_p)^2} \\ &\precsim \max_{p \in [m]} |\tilde{\alpha}_p - \alpha_p| = \|\tilde{\boldsymbol{\alpha}} - \boldsymbol{\alpha}\|_\infty \precsim \varepsilon. \end{aligned} \tag{15}$$

Notice that $\widetilde{\mathbf{H}}$ is the first $k$ eigenvectors of $\frac{1}{ns}\mathbf{P}_{\tilde{\boldsymbol{\alpha}}}\mathbf{P}_{\tilde{\boldsymbol{\alpha}}}^\top$. Then, by Lemma C.1,

$$\left\| \widetilde{\mathbf{H}}\widetilde{\mathbf{H}}^\top - \overline{\mathbf{H}}\,\overline{\mathbf{H}}^\top \right\|_{\mathrm{F}} \precsim \frac{\sqrt{k} \left\| \frac{1}{ns}\mathbf{P}_{\tilde{\boldsymbol{\alpha}}}\mathbf{P}_{\tilde{\boldsymbol{\alpha}}}^\top - \frac{1}{n^2}\mathbf{K}_{\tilde{\boldsymbol{\alpha}}}^2 \right\|}{\delta(\tilde{\boldsymbol{\alpha}})} \precsim \sqrt{k} \left\| \frac{1}{ns}\mathbf{P}_{\tilde{\boldsymbol{\alpha}}}\mathbf{P}_{\tilde{\boldsymbol{\alpha}}}^\top - \frac{1}{n^2}\mathbf{K}_{\tilde{\boldsymbol{\alpha}}}^2 \right\|. \tag{16}$$

For any kernel matrix $\mathbf{K} \in \mathbb{R}^{n \times n}$, let $\mathbf{P} \in \mathbb{R}^{n \times s}$ be its $s$ columns selected by uniform sampling. Let $\psi_i = \frac{1}{\sqrt{n}}\boldsymbol{\Phi}_n^\top \phi(x_i)$ in Theorem B.3. Then, in Theorem B.3, $\frac{1}{n}\boldsymbol{\Psi}_n\boldsymbol{\Psi}_n^\top = \frac{1}{n^2}\mathbf{K}^2$ and $\frac{1}{ns}\boldsymbol{\Psi}_I\boldsymbol{\Psi}_I^\top = \frac{1}{ns}\mathbf{P}\mathbf{P}^\top$. By Theorem B.3, with probability at least $1 - \delta$,

$$\left\| \frac{1}{ns}\mathbf{P}\mathbf{P}^\top - \frac{1}{n^2}\mathbf{K}^2 \right\| \precsim \varepsilon.$$

According to Eq.(16), we have $\left\| \widetilde{\mathbf{H}}\widetilde{\mathbf{H}}^\top - \overline{\mathbf{H}}\,\overline{\mathbf{H}}^\top \right\|_{\mathrm{F}} \precsim \sqrt{k}\varepsilon$.

Combining Eq.(15), with probability at least $1 - \delta$,

$$\left\| \widetilde{\mathbf{H}}\widetilde{\mathbf{H}}^\top - \mathbf{H}\mathbf{H}^\top \right\|_{\mathrm{F}} \leq \left\| \widetilde{\mathbf{H}}\widetilde{\mathbf{H}}^\top - \overline{\mathbf{H}\mathbf{H}}^\top \right\|_{\mathrm{F}} + \left\| \overline{\mathbf{H}\mathbf{H}}^\top - \mathbf{H}\mathbf{H}^\top \right\|_{\mathrm{F}} \precsim \varepsilon.$$

$\square$

## C.2. Proof of Theorem 4.1

*Proof.* Denote that $\mathbf{T} = [\mathbf{t}_1, \cdots, \mathbf{t}_s]$. Then, $\mathbf{A}_j = \frac{1}{ns}\mathbf{K}\mathbf{t}_j\mathbf{t}_j^\top\mathbf{K}$ and $\sum_{j=1}^s \mathbf{A}_j = \frac{1}{ns}\mathbf{K}\mathbf{T}\mathbf{T}^\top\mathbf{K}$. It is can be checked that $\mathbb{E}[\mathbf{K}\mathbf{t}_j\mathbf{t}_j^\top\mathbf{K}] = \frac{1}{n}\mathbf{K}^2$. Thus, $\sum_{j=1}^s \mathbb{E}[\mathbf{A}_j] = \frac{1}{n^2}\mathbf{K}^2$. Moreover, $\|\mathbf{A}_j\| = \frac{1}{ns}\|\mathbf{K}\mathbf{t}_j\mathbf{t}_j^\top\mathbf{K}\| \leq \frac{1}{s}$.

By Theorem B.2, for any $\delta > 0$, we have

$$\Pr\left[\lambda_t(\sum_j \mathbf{A}_j) \geq (1+\delta)\mu_t\right] \leq (n-k+1) \cdot \left[e^{\delta-(1+\delta)\log(1+\delta)}\right]^{s\mu_t} \leq (n-t+1) \cdot e^{-\frac{s\delta^2\mu_k}{2}} \tag{17}$$

(Because $\delta - (1+\delta)\log(1+\delta) \leq -\delta^2/2$.)

Let $\delta = \frac{\varepsilon}{\sqrt{\mu_t}}$, we have

$$\Pr\left[\lambda_t(\sum_j \mathbf{A}_j) \geq (1+\delta)\mu_t\right] \leq (n-t+1) \cdot e^{-\frac{s\varepsilon^2}{2}}.$$

Consequently, with probability at least $1 - (n-t+1) \cdot e^{-\frac{s\varepsilon^2}{2}}$,

$$\lambda_t(\sum_j \mathbf{A}_j) \leq (1 + \frac{\varepsilon}{\sqrt{\mu_t}})\mu_t.$$

Thus, for any $s \geq \frac{2\log(n/\delta)}{\varepsilon^2}$,

$$\sqrt{\lambda_t(\sum_j \mathbf{A}_j)} \leq \sqrt{1 + \frac{\varepsilon}{\sqrt{\mu_t}}} \cdot \sqrt{\mu_t} \leq (1 + \frac{\varepsilon}{\sqrt{\mu_t}})\sqrt{\mu_t}.$$

By the definition of $\sigma_t\left(\frac{1}{\sqrt{ns}}\right)$ and $\mu_t$, we have

$$\sigma_t\left(\frac{1}{\sqrt{ns}}\mathbf{P}\right) \leq \lambda_t\left(\frac{1}{n}\mathbf{K}\right) + \varepsilon. \tag{18}$$

Now, we proceed to prove the other half of Theorem 4.1. According to Theorem B.2, for any $\delta \in [0, 1)$,

$$\Pr\left[\lambda_t(\sum_j \mathbf{A}_j) \leq (1-\delta)\mu_t\right] \leq t \cdot \left[e^{-\delta-(1-\delta)\log(1-\delta)}\right]^{s\mu_t} \leq t \cdot e^{-\frac{s\delta^2\mu_k}{2}} \tag{19}$$

(Because $\delta + (1-\delta)\log(1-\delta) \geq \delta^2/2$.)

Let $\delta = \frac{\varepsilon}{\sqrt{\mu_t}}$, we have

$$\Pr\left[\lambda_t(\sum_j \mathbf{A}_j) \leq (1-\delta)\mu_t\right] \leq t \cdot e^{-\frac{s\varepsilon^2}{2}},$$

which is equivalent to

$$(1 - \frac{\varepsilon}{\sqrt{\mu_t}})\mu_t \leq \lambda_k(\sum_j \mathbf{A}_j)$$

holds with probability at least $1 - t \cdot e^{-\frac{s\varepsilon^2}{2}}$. For any $s \geq \frac{2\log(n/\delta)}{\varepsilon^2}$, due to $\sqrt{1 - \frac{\varepsilon}{\sqrt{\mu_t}}} \geq 1 - \frac{\varepsilon}{\sqrt{\mu_t}}$, we have

$$(1 - \frac{\varepsilon}{\sqrt{\mu_t}})\sqrt{\mu_t} \leq \sqrt{(1 - \frac{\varepsilon}{\sqrt{\mu_t}})\mu_t} \leq \sqrt{\lambda_t(\sum_j \mathbf{A}_j)},$$

holds with probability at least $1 - \delta$. By the definition of $\sigma_t\left(\frac{1}{\sqrt{ns}}\right)$ and $\mu_t$, we have

$$\lambda_t\left(\frac{1}{n}\mathbf{K}\right) - \varepsilon \leq \sigma_t\left(\frac{1}{\sqrt{ns}}\mathbf{P}\right). \tag{20}$$

Combining Eq.(18) and Eq.(20), by union bound, when $s \geq \frac{2\log(2n/\delta)}{\varepsilon^2}$, with probability at least $1 - \delta$,

$$\left|\sigma_t\left(\frac{1}{\sqrt{ns}}\mathbf{P}\right) - \lambda_t\left(\frac{1}{n}\mathbf{K}\right)\right| \leq \varepsilon.$$

$\square$

## C.3. Proof of Lemma 4.2 and Lemma 4.3

**Lemma C.2** (Theorem 7.3.2, (Songgui et al., 2006)). *Assume that* $\mathbf{A}, \mathbf{B}$ *are two PSD matrices, and* $\mathbf{A}^2 \preccurlyeq \mathbf{B}^2$. *Then,* $\mathbf{A} \preccurlyeq \mathbf{B}$.

**Lemma C.3.** *If* $s \geq c\log(n/\delta)/\varepsilon^2$, *with probability at least* $1 - \delta$,

$$\left\|\frac{1}{s}\mathbf{\Phi}_s^\top \mathbf{\Phi}_s - \frac{1}{\sqrt{ns}}(\mathbf{\Phi}_s^\top \mathbf{\Phi}_n \mathbf{\Phi}_n^\top \mathbf{\Phi}_s)^{1/2}\right\| \leq \varepsilon.$$

*Proof.* Assume that $\psi_i = \left(\frac{1}{s}\mathbf{\Phi}_s^\top \mathbf{\Phi}_s\right)^+ \left(\frac{1}{\sqrt{s}}\mathbf{\Phi}_s^\top \phi(x_i)\right)$. Then, it is easy to check that there exists a constant $c > 0$ such that $\|\psi_i\| \leq c$ and $\left\|\frac{1}{n}\mathbf{\Psi}_n\mathbf{\Psi}_n^\top\right\| \leq c$. By Theorem B.3, we have

$$\Pr\left[\left\|\frac{1}{n}\mathbf{\Psi}_n\mathbf{\Psi}_n^\top - \frac{1}{s}\mathbf{\Psi}_I\mathbf{\Psi}_I^\top\right\| > \varepsilon\right] \leq n\exp\left(\frac{-s\varepsilon^2/2}{c\cdot(c+t/3)}\right) \leq n\exp\left(\frac{-s\varepsilon^2}{4c^2}\right).$$

It is equivalent to, for all $s \geq c\log(n/\delta)/\varepsilon^2$, with probability at least $1 - \delta$,

$$\left\|\left(\frac{1}{s}\mathbf{\Phi}_s^\top \mathbf{\Phi}_s\right)^+ \left(\frac{1}{ns}\mathbf{\Phi}_s^\top \mathbf{\Phi}_n\mathbf{\Phi}_n^\top \mathbf{\Phi}_s - \frac{1}{s^2}\mathbf{\Phi}_s^\top \mathbf{\Phi}_s\mathbf{\Phi}_s^\top \mathbf{\Phi}_s\right) \left(\frac{1}{s}\mathbf{\Phi}_s^\top \mathbf{\Phi}_s\right)^+\right\| \leq \varepsilon,$$

which implies

$$(1 - \varepsilon) \cdot \left(\frac{1}{s^2}\mathbf{\Phi}_s^\top \mathbf{\Phi}_s\mathbf{\Phi}_s^\top \mathbf{\Phi}_s\right) \preccurlyeq \frac{1}{ns}\mathbf{\Phi}_s^\top \mathbf{\Phi}_n\mathbf{\Phi}_n^\top \mathbf{\Phi}_s \preccurlyeq (1 + \varepsilon) \cdot \left(\frac{1}{s^2}\mathbf{\Phi}_s^\top \mathbf{\Phi}_s\mathbf{\Phi}_s^\top \mathbf{\Phi}_s\right).$$

By Lemma C.2, we have

$$\sqrt{1 - \varepsilon} \cdot \left(\frac{1}{s}\mathbf{\Phi}_s^\top \mathbf{\Phi}_s\right) \preccurlyeq \left(\frac{1}{ns}\mathbf{\Phi}_s^\top \mathbf{\Phi}_n\mathbf{\Phi}_n^\top \mathbf{\Phi}_s\right)^{1/2} \preccurlyeq \sqrt{1 + \varepsilon} \cdot \left(\frac{1}{n}\mathbf{\Phi}_s^\top \mathbf{\Phi}_s\right).$$

For any $\varepsilon \in (0, 1)$, due to $1 - \varepsilon \leq \sqrt{1 - \varepsilon}$ and $\sqrt{1 + \varepsilon} \leq 1 + \varepsilon$, we have

$$(1 - \varepsilon) \cdot \left(\frac{1}{s}\mathbf{\Phi}_s^\top \mathbf{\Phi}_s\right) \preccurlyeq \left(\frac{1}{ns}\mathbf{\Phi}_s^\top \mathbf{\Phi}_n\mathbf{\Phi}_n^\top \mathbf{\Phi}_s\right)^{1/2} \preccurlyeq (1 + \varepsilon) \cdot \left(\frac{1}{s}\mathbf{\Phi}_s^\top \mathbf{\Phi}_s\right)$$

Then, we can obtain

$$\left\| \left( \frac{1}{ns} \mathbf{\Phi}_s^\top \mathbf{\Phi}_n \mathbf{\Phi}_n^\top \mathbf{\Phi}_s \right)^{1/2} - \frac{1}{s} \mathbf{\Phi}_s^\top \mathbf{\Phi}_s \right\| \leq \varepsilon \left\| \frac{1}{s} \mathbf{\Phi}_s^\top \mathbf{\Phi}_s \right\| \precsim \varepsilon.$$

$\square$

*Proof of Lemma 4.2.* For convenience of expression, we use $\{\hat{h}_j\}_{j=1}^k$ to present $\{\hat{h}_j^\gamma\}_{j=1}^k$ and $\{\tilde{h}_j\}_{j=1}^k$ to to present $\{\tilde{h}_j^\gamma\}_{j=1}^k$. Then, by triangle inequality, we have

$$|\mathcal{T}_n(K_p, \{\hat{h}_j\}_{j=1}^k) - \widetilde{\mathcal{T}}_s(\widetilde{K}_p, \{\tilde{h}_j\}_{j=1}^k)|$$

$$\leq \sum_{j=1}^k \left| \frac{1}{n^2} \sum_{i=1}^n \sum_{t=1}^n K_p(x_i, x_t)\hat{h}_j(x_i)\hat{h}_j(x_t) - \frac{1}{s^2} \sum_{i=1}^s \sum_{t=1}^s \widetilde{K}_p(a_i, a_t)\tilde{h}_j(a_i)\tilde{h}_j(a_t) \right|$$

$$\leq \sum_{j=1}^k \underbrace{\left| \frac{1}{n^2} \sum_{i=1}^n \sum_{t=1}^n K_p(x_i, x_t)\hat{h}_j(x_i)\hat{h}_j(x_t) - \frac{1}{s^2} \sum_{i=1}^s \sum_{t=1}^s K_p(a_i, a_t)\hat{h}_j(a_i)\hat{h}_j(a_t) \right|}_{\mathcal{A}}$$

$$+ \sum_{j=1}^k \underbrace{\left| \frac{1}{s^2} \sum_{i=1}^s \sum_{t=1}^s K_p(a_i, a_t)\hat{h}_j(a_i)\hat{h}_j(a_t) - \frac{1}{s^2} \sum_{i=1}^s \sum_{t=1}^s \widetilde{K}_p(a_i, a_t)\hat{h}_j(a_i)\hat{h}_j(a_t) \right|}_{\mathcal{B}}$$

$$+ \sum_{j=1}^k \underbrace{\left| \frac{1}{s^2} \sum_{i=1}^s \sum_{t=1}^s \widetilde{K}_p(a_i, a_t)\hat{h}_j(a_i)\hat{h}_j(a_t) - \frac{1}{s^2} \sum_{i=1}^s \sum_{t=1}^s \widetilde{K}_p(a_i, a_t)\tilde{h}_j(a_i)\tilde{h}_j(a_t) \right|}_{\mathcal{C}} \quad (21)$$

For any $x \in \mathcal{X}$, assume that $\boldsymbol{\psi}(x) = \hat{h}_j(x)\boldsymbol{\phi}_p(x)$. For all $i \in [n]$, it is easy to check that $\|\boldsymbol{\psi}(x_i)\| \leq c$ and $\mathbb{E}[\|\boldsymbol{\psi}(x_i)\|^2] \leq c$ with some constant $c > 0$. By Theorem B.4, with probability at least $1 - \delta$,

$$\left\| \frac{1}{s} \sum_{i=1}^s \boldsymbol{\psi}(a_i) - \frac{1}{n} \sum_{i=1}^n \boldsymbol{\psi}(x_i) \right\| \leq \varepsilon.$$

For Item $\mathcal{A}$ in Eq. (21), with probability at least $1 - \delta$,

$$\mathcal{A} = \left| \left\| \frac{1}{n} \sum_{i=1}^n \boldsymbol{\psi}(x_i) \right\|^2 - \left\| \frac{1}{s} \sum_{i=1}^s \boldsymbol{\psi}(a_i) \right\|^2 \right| \leq \left\| \frac{1}{n} \sum_{i=1}^n \boldsymbol{\psi}(x_i) - \frac{1}{s} \sum_{i=1}^s \boldsymbol{\psi}(a_i) \right\| \cdot \left\| \frac{1}{n} \sum_{i=1}^n \boldsymbol{\psi}(x_i) + \frac{1}{s} \sum_{i=1}^s \boldsymbol{\psi}(a_i) \right\| \precsim \varepsilon. \quad (22)$$

For Item $\mathcal{B}$ in Eq. (21), according to Lemma C.3, with probability at least $1 - \delta$,

$$\mathcal{B} \precsim \left\| \frac{1}{s} \mathbf{K}_p - \frac{1}{s} \widetilde{\mathbf{K}}_p \right\| = \left\| \frac{1}{s} \mathbf{\Phi}_s^\top \mathbf{\Phi}_s - \frac{1}{\sqrt{ns}} (\mathbf{\Phi}_s^\top \mathbf{\Phi}_n \mathbf{\Phi}_n^\top \mathbf{\Phi}_s)^{1/2} \right\| \leq \varepsilon. \quad (23)$$

For Item $\mathcal{C}$ in Eq. (21), we have

$$\mathcal{C} \leq \frac{1}{s^2} \sum_{i=1}^s \sum_{t=1}^s |\widetilde{K}_p(a_i, a_t)| \cdot |\hat{h}_j(a_i)\hat{h}_j(a_t) - \tilde{h}_j(a_i)\tilde{h}_j(a_t)|$$

$$\precsim \sup_{x,y} |t_j\hat{h}_j(x) \cdot t_j\hat{h}_j(y) - \tilde{h}_j(x)\tilde{h}_j(y)|$$

$$\leq \sup_{x,y} |t_j\hat{h}_j(x) \cdot a_j\hat{h}_j(y) - t_j\hat{h}_j(x)\tilde{h}_j(y) + a_j\hat{h}_j(x)\tilde{h}_j(y) - \tilde{h}_j(x)\tilde{h}_j(y)| \quad (24)$$

$$\leq \sup_{x,y} |a_j\hat{h}_j(x)| \cdot |a_j\hat{h}_j(y) - \tilde{h}_j(y)| + \sup_{x,y} |\tilde{h}_j(y)| \cdot |a_j\hat{h}_j(x) - \tilde{h}_j(x)|$$

$$\precsim \|a_j\hat{h}_j - \tilde{h}_j\|_\infty.$$

For any $x, y \in \mathcal{X}$, denote that

$$\hat{K}(x,y) = \frac{1}{n}\sum_{i=1}^{n} K(x,x_i)K(x_i,y), \ L_{\hat{K}}f(x) = \frac{1}{n^2}\sum_{i=1}^{n}\sum_{t=1}^{n} K(x,x_i)K(x_i,x_t)f(x_t),$$

and

$$\overline{K}(x,y) = \frac{1}{n}\sum_{i=1}^{n} K(x,x_i)K(x_i,y), \ L_{\overline{K}}f(x) = \frac{1}{ns}\sum_{i=1}^{n}\sum_{t=1}^{s} K(x,x_i)K(x_i,a_t)f(a_t).$$

By the definitions of eigenfunctions, it can be checked that $\{\hat{h}_j\}_{j=1}^{k}$ and $\{\tilde{h}_j\}_{j=1}^{k}$ are $\{\tilde{h}_j\}_{j=1}^{k}$ the eigenfunctions of $L_{\hat{K}}$ and $L_{\overline{K}}$, respectively. According to Proposition 18 of (Von Luxburg et al., 2008) and Lemma B.1, we have

$$\|a_j\hat{h}_j - \tilde{h}_j\|_{\infty} \precsim \|(L_{\hat{K}} - L_{\overline{K}})h_j\|_{\infty} + \|(L_{\hat{K}} - L_{\overline{K}})L_{\overline{K}}\|) \precsim \|L_{\hat{K}} - L_{\overline{K}}\|. \tag{25}$$

Then, we process to magnify $\|L_{\hat{K}} - L_{\overline{K}}\|$.

$$
\begin{aligned}
\|L_{\hat{K}} - L_{\overline{K}}\| &= \sup_{\substack{x \in \mathcal{X} \\ \|f\|_\infty = 1}} \left| \frac{1}{n^2}\sum_{i=1}^{n}\sum_{t=1}^{n} K(x,x_i)K(x_i,x_t)f(x_t) - \frac{1}{ns}\sum_{i=1}^{n}\sum_{t=1}^{s} K(x,x_i)K(x_i,a_t)f(a_t) \right| \\
&= \sup_{\substack{x \in \mathcal{X} \\ \|f\|_\infty = 1}} \left| \frac{1}{n^2}\sum_{i=1}^{n}\sum_{t=1}^{n} K(x,x_i)K(x_i,x_t)f(x_t) - \frac{1}{ns}\sum_{i=1}^{n}\sum_{t=1}^{s} K(x,x_i)K(x_i,a_t)f(a_t) \right| \\
&\leq \sup_{\substack{x \in \mathcal{X} \\ \|f\|_\infty = 1}} \frac{1}{n}\sum_{i=1}^{n}\left( |K(x,x_i)| \cdot \left| \frac{1}{n}\sum_{t=1}^{n} K(x_i,x_t)f(x_t) - \frac{1}{s}\sum_{t=1}^{s} K(x_i,a_t)f(a_t) \right| \right) \\
&\precsim \sup_{\|f\|_\infty = 1} \frac{1}{n}\sum_{i=1}^{n} \left| \left\langle \phi(x_i), \frac{1}{n}\sum_{t=1}^{n} f(x_t)\phi(x_t) - \frac{1}{s}\sum_{t=1}^{s} f(a_t)\phi(a_t) \right\rangle \right| \\
&\leq \sup_{\|f\|_\infty = 1} \frac{1}{n}\sum_{i=1}^{n} \|\phi(x_i)\| \cdot \left\| \frac{1}{n}\sum_{t=1}^{n} f(x_t)\phi(x_t) - \frac{1}{s}\sum_{t=1}^{s} f(a_t)\phi(a_t) \right\| \\
&\leq \frac{1}{n}\sum_{i=1}^{n} \|\phi(x_i)\| \cdot \varepsilon \precsim \varepsilon. \ (\text{By Lemma B.4.})
\end{aligned}
\tag{26}
$$

Combining Eq.(24), Eq.(25) and Eq.(26), we know that $\mathcal{C} \precsim \varepsilon$. According the derived bounds for $\mathcal{A}$ and $\mathcal{B}$, if $s \geq c\log(n/\delta)/\varepsilon^2$, with probability at least $1 - \delta$,

$$|\mathcal{T}_n(K_p, \{\hat{h}_j\}) - \widetilde{\mathcal{T}}_s(\widetilde{K}_p, \{\tilde{h}_j\})| \leq k\varepsilon.$$

$\square$

*Proof of Lemma 4.3.* For any $\boldsymbol{\alpha}, \boldsymbol{\beta} \in \Delta$, denote that $\mathbf{H}_{\boldsymbol{\alpha}}, \mathbf{H}_{\boldsymbol{\beta}}$ are composed of the first $k$ eigenvectors of $\mathbf{K}_{\boldsymbol{\alpha}}$ and $\mathbf{K}_{\boldsymbol{\beta}}$, respectively. Then, for any orthogonal matrix $\hat{\mathbf{O}} \in \mathbb{R}^{k \times k}$, we have

$$
\begin{aligned}
&|\mathcal{T}_n(K_p, \{\hat{h}_j^{\boldsymbol{\alpha}}\}_{j=1}^{k}) - \mathcal{T}_n(K_p, \{\hat{h}_j^{\boldsymbol{\beta}}\}_{j=1}^{k})| \\
&= \left| \frac{1}{n}\text{tr}(\mathbf{K}_p\mathbf{H}_{\boldsymbol{\alpha}}\mathbf{H}_{\boldsymbol{\alpha}}^{\top}) - \frac{1}{n}\text{tr}(\mathbf{K}_p\mathbf{H}_{\boldsymbol{\beta}}\mathbf{H}_{\boldsymbol{\beta}}^{\top}) \right| \\
&\leq \left\| \frac{\mathbf{K}_p}{n} \right\|_{\text{F}} \cdot \left\| \mathbf{H}_{\boldsymbol{\alpha}}\mathbf{H}_{\boldsymbol{\alpha}}^{\top} - \mathbf{H}_{\boldsymbol{\beta}}\mathbf{H}_{\boldsymbol{\beta}}^{\top} \right\|_{\text{F}} \\
&\leq \left\| \mathbf{H}_{\boldsymbol{\alpha}}\hat{\mathbf{O}}\hat{\mathbf{O}}^{\top}\mathbf{H}_{\boldsymbol{\alpha}}^{\top} - \mathbf{H}_{\boldsymbol{\alpha}}\hat{\mathbf{O}}\mathbf{H}_{\boldsymbol{\beta}}^{\top} \right\|_{\text{F}} + \left\| \mathbf{H}_{\boldsymbol{\alpha}}\hat{\mathbf{O}}\mathbf{H}_{\boldsymbol{\beta}}^{\top} - \mathbf{H}_{\boldsymbol{\beta}}\mathbf{H}_{\boldsymbol{\beta}}^{\top} \right\|_{\text{F}} \\
&\leq \|\mathbf{H}_{\boldsymbol{\alpha}}\hat{\mathbf{O}}\| \cdot \left\| \mathbf{H}_{\boldsymbol{\alpha}}\hat{\mathbf{O}} - \mathbf{H}_{\boldsymbol{\beta}} \right\|_{\text{F}} + \|\mathbf{H}_{\boldsymbol{\beta}}\| \cdot \left\| \mathbf{H}_{\boldsymbol{\alpha}}\hat{\mathbf{O}} - \mathbf{H}_{\boldsymbol{\beta}} \right\|_{\text{F}} \\
&\leq 2 \left\| \mathbf{H}_{\boldsymbol{\alpha}}\hat{\mathbf{O}} - \mathbf{H}_{\boldsymbol{\beta}} \right\|_{\text{F}}.
\end{aligned}
\tag{27}
$$

For any vector $\boldsymbol{\alpha} \in \mathbb{R}^m$, let $\delta(\boldsymbol{\alpha})$ denote the gap between the $k$-th and $(k+1)$-th eigenvalues of the matrix $\frac{1}{n}\mathbf{K}_{\boldsymbol{\alpha}}$. By Assumption 3.2, there exists a constant $c \geq 0$ such that for any $\boldsymbol{\alpha} \in \triangle$, $\delta(\boldsymbol{\gamma}) \geqslant 1/c$. Using Lemma C.1, let $r = 1$ and $s = k$, then we have:

$$\left\| \mathbf{H}_{\boldsymbol{\alpha}} \hat{\mathbf{O}} - \mathbf{H}_{\boldsymbol{\beta}} \right\|_{\mathrm{F}} \precsim \frac{\left\| \frac{1}{n}\mathbf{K}_{\boldsymbol{\alpha}} - \frac{1}{n}\mathbf{K}_{\boldsymbol{\beta}} \right\|_{\mathrm{F}}}{\delta(\boldsymbol{\alpha})} \precsim \|\boldsymbol{\alpha} - \boldsymbol{\beta}\|_{\infty}. \tag{28}$$

Combining Eq.(27) and Eq.(28),

$$|\mathcal{T}_n(K_p, \{\hat{h}_j^{\boldsymbol{\alpha}}\}_{j=1}^k) - \mathcal{T}_n(K_p, \{\hat{h}_j^{\boldsymbol{\beta}}\}_{j=1}^k)| \precsim \|\boldsymbol{\alpha} - \boldsymbol{\beta}\|_{\infty}.$$

Thus, according to Lemma 4.2, with probability at least $1 - \delta$,

$$\begin{aligned}
&|\mathcal{T}_n(K_p, \{\hat{h}_j^{\boldsymbol{\alpha}}\}_{j=1}^k) - \mathcal{T}_s(\widetilde{K}_p, \{\tilde{h}_j^{\boldsymbol{\beta}}\}_{j=1}^k)| \\
\leq& |\mathcal{T}_n(K_p, \{\hat{h}_j^{\boldsymbol{\alpha}}\}_{j=1}^k) - \mathcal{T}_n(K_p, \{\hat{h}_j^{\boldsymbol{\beta}}\}_{j=1}^k)| + |\mathcal{T}_n(K_p, \{\hat{h}_j^{\boldsymbol{\beta}}\}_{j=1}^k) - \mathcal{T}_s(\widetilde{K}_p, \{\tilde{h}_j^{\boldsymbol{\beta}}\}_{j=1}^k)| \\
\leq& \|\boldsymbol{\alpha} - \boldsymbol{\beta}\|_{\infty} + k\varepsilon.
\end{aligned} \tag{29}$$

$\square$

### C.4. Proof of Theorem 4.4

*Proof.* **1) Proof for SMKKM.** When the input is original base kernel matrices, in the updating process, we assume that the kernel weights are $\boldsymbol{\alpha}^{(0)}, \cdots, \boldsymbol{\alpha}^{(t)}, \cdots, \boldsymbol{\alpha}^{(T)}$, in which $\boldsymbol{\alpha}^{(t)}$ denotes the kernel weights after the $t$-th updating. Correspondingly, when the input is core kernel matrices, assume that the kernel weights are $\boldsymbol{\beta}^{(0)}, \cdots, \boldsymbol{\beta}^{(t)}, \cdots, \boldsymbol{\beta}^{(T)}$ in the optimization process. By the assumption of the same initialization of kernel weights, we have $\boldsymbol{\alpha}^{(0)} = \boldsymbol{\beta}^{(0)}$.

With some fixed index $u \in [m]$, for the $t$-th step, according to Lemma 4.3, we have

$$\begin{aligned}
&|\alpha_u^{(t+1)} - \beta_u^{(t+1)}| - |\alpha_u^{(t)} - \beta_u^{(t)}| \\
\leq& |\alpha_u^{(t+1)} - \alpha_u^{(t)} - (\beta_u^{(t+1)} - \beta_u^{(t)})| \\
\leq& \frac{1}{m-1} \left| \sum_{p \neq u} \left( \alpha_p^{(t)} \mathcal{T}_n(K_p, \{\hat{h}_j^{\boldsymbol{\alpha}^{(t)}}\}_{j=1}^k) - \alpha_u^{(t)} \mathcal{T}_n(K_u, \{\hat{h}_j^{\boldsymbol{\alpha}^{(t)}}\}_{j=1}^k) \right) \right. \\
&\left. - \sum_{p \neq u} \left( \beta_p^{(t)} \mathcal{T}_s(\widetilde{K}_p, \{\tilde{h}_j^{\boldsymbol{\beta}^{(t)}}\}_{j=1}^k) - \beta_u^{(t)} \mathcal{T}(\widetilde{K}_u, \{\tilde{h}_j^{\boldsymbol{\beta}^{(t)}}\}_{j=1}^k) \right) \right| \\
\precsim& \max_{q \in [m]} \left| \alpha_q^{(t)} \mathcal{T}_n(K_q, \{\hat{h}_j^{\boldsymbol{\alpha}^{(t)}}\}_{j=1}^k) - \beta_q^{(t)} \mathcal{T}_s(\widetilde{K}_q, \{\tilde{h}_j^{\boldsymbol{\beta}^{(t)}}\}_{j=1}^k) \right| \\
=& \max_{q \in [m]} \left| \alpha_q^{(t)} \mathcal{T}_n(K_q, \{\hat{h}_j^{\boldsymbol{\alpha}^{(t)}}\}_{j=1}^k) - \beta_q^{(t)} \mathcal{T}_n(K_q, \{\hat{h}_j^{\boldsymbol{\alpha}^{(t)}}\}_{j=1}^k) + \beta_q^{(t)} \mathcal{T}_n(K_q, \{\hat{h}_j^{\boldsymbol{\alpha}^{(t)}}\}_{j=1}^k) - \beta_q^{(t)} \mathcal{T}_s(\widetilde{K}_q, \{\tilde{h}_j^{\boldsymbol{\beta}^{(t)}}\}_{j=1}^k) \right| \\
\leq& \max_{q \in [m]} |\alpha_q^{(t)} - \beta_q^{(t)}| \cdot \mathcal{T}_n(K_q, \{\hat{h}_j^{\boldsymbol{\alpha}^{(t)}}\}_{j=1}^k) + \beta_q^{(t)} \cdot \left| \mathcal{T}_n(K_q, \{\hat{h}_j^{\boldsymbol{\alpha}^{(t)}}\}_{j=1}^k) - \mathcal{T}_s(\widetilde{K}_q, \{\tilde{h}_j^{\boldsymbol{\beta}^{(t)}}\}_{j=1}^k) \right| \\
\precsim& \max_{q \in [m]} |\alpha_q^{(t)} - \beta_q^{(t)}| + \|\boldsymbol{\alpha}^{(t)} - \boldsymbol{\beta}^{(t)}\|_{\infty} + k\varepsilon \\
\precsim& \|\boldsymbol{\alpha}^{(t)} - \boldsymbol{\beta}^{(t)}\|_{\infty} + k\varepsilon.
\end{aligned} \tag{30}$$

Similarly, for $p \in [m], p \neq u$, we have

$$\begin{aligned}
&|\alpha_p^{(t+1)} - \beta_p^{(t+1)}| - |\alpha_p^{(t)} - \beta_p^{(t)}| \\
\leq& |\alpha_p^{(t+1)} - \alpha_p^{(t)} - (\beta_p^{(t+1)} - \beta_p^{(t)})| \\
\leq& \frac{1}{m-1} \left| \alpha_u^{(t)} \mathcal{T}_n(K_u, \{\hat{h}_j^{\boldsymbol{\alpha}^{(t)}}\}_{j=1}^k) - \alpha_p^{(t)} \mathcal{T}_n(K_p, \{\hat{h}_j^{\boldsymbol{\alpha}^{(t)}}\}_{j=1}^k) \right| \\
&+ \frac{1}{m-1} \left| \beta_u^{(t)} \mathcal{T}_s(\widetilde{K}_u, \{\tilde{h}_j^{\boldsymbol{\beta}^{(t)}}\}_{j=1}^k) - \beta_p^{(t)} \mathcal{T}_s(\widetilde{K}_p, \{\tilde{h}_j^{\boldsymbol{\beta}^{(t)}}\}_{j=1}^k) \right| \\
\precsim& \|\boldsymbol{\alpha}^{(t)} - \boldsymbol{\beta}^{(t)}\|_{\infty} + k\varepsilon.
\end{aligned} \tag{31}$$

Combining Eq.(30) and Eq.(31), with probability at least $1 - \delta$,

$$\|\boldsymbol{\alpha}^{(t+1)} - \boldsymbol{\beta}^{(t+1)}\|_\infty \precsim \|\boldsymbol{\alpha}^{(t)} - \boldsymbol{\beta}^{(t)}\|_\infty + k\varepsilon.$$

Based on the above recurrence relation, it can be concluded that if $s \geq c\log(nT/\delta)/\varepsilon^2 = \widetilde{\mathcal{O}}(1/\varepsilon^2)$, with probability at least $1 - \delta$,

$$\|\boldsymbol{\alpha}^{(T)} - \boldsymbol{\beta}^{(T)}\|_\infty \precsim \|\boldsymbol{\alpha}^{(T-1)} - \boldsymbol{\beta}^{(T-1)}\|_\infty + k\varepsilon \precsim \cdots \precsim \|\boldsymbol{\alpha}^{(0)} - \boldsymbol{\beta}^{(0)}\|_\infty + k\varepsilon,$$

which satisfies the condition of Definition 3.1.

**2) Proof for SMKKM-KWR.** The proof for SMKKM-KWR is similar to SMKKM. We use the same notation to represent the kernel weight changes at each iteration step.

With some fixed index $u \in [m]$, for the $t$-th step, according to Lemma 4.3, we have

$$
\begin{aligned}
&|\alpha_u^{(t+1)} - \beta_u^{(t+1)}| - |\alpha_u^{(t)} - \beta_u^{(t)}| \\
\leq &|\alpha_u^{(t+1)} - \alpha_u^{(t)} - (\beta_u^{(t+1)} - \beta_u^{(t)})| \\
\leq &\frac{1}{m-1}\left| \sum_{p \neq u} \left( \alpha_p^{(t)}(\mathcal{T}_n(K_p, \{\hat{h}_j^{\boldsymbol{\alpha}^{(t)}}\}_{j=1}^k) + \lambda) - \alpha_u^{(t)}(\mathcal{T}_n(K_u, \{\hat{h}_j^{\boldsymbol{\alpha}^{(t)}}\}_{j=1}^k) + \lambda) \right) \right. \\
&\left. - \sum_{p \neq u} \left( \beta_p^{(t)}(\mathcal{T}_s(\widetilde{K}_p, \{\tilde{h}_j^{\boldsymbol{\beta}^{(t)}}\}_{j=1}^k) + \lambda) - \beta_u^{(t)}(\mathcal{T}(\widetilde{K}_u, \{\tilde{h}_j^{\boldsymbol{\beta}^{(t)}}\}_{j=1}^k) + \lambda) \right) \right| \\
\precsim &\max_{q \in [m]} \left| \alpha_q^{(t)}(\mathcal{T}_n(K_q, \{\hat{h}_j^{\boldsymbol{\alpha}^{(t)}}\}_{j=1}^k) + \lambda) - \beta_q^{(t)}(\mathcal{T}_s(\widetilde{K}_q, \{\tilde{h}_j^{\boldsymbol{\beta}^{(t)}}\}_{j=1}^k) + \lambda) \right| \\
\leq &\max_{q \in [m]} \left| \alpha_q^{(t)}\mathcal{T}_n(K_q, \{\hat{h}_j^{\boldsymbol{\alpha}^{(t)}}\}_{j=1}^k) - \beta_q^{(t)}\mathcal{T}_s(\widetilde{K}_q, \{\tilde{h}_j^{\boldsymbol{\beta}^{(t)}}\}_{j=1}^k) \right| + \lambda|\alpha_q^{(t)} - \beta_q^{(t)}| \\
\precsim &\lambda\|\boldsymbol{\alpha}^{(t)} - \boldsymbol{\beta}^{(t)}\|_\infty + k\varepsilon.
\end{aligned}
\tag{32}
$$

Similar, for $p \neq u, p \in [m]$,

$$|\alpha_p^{(t+1)} - \beta_p^{(t+1)}| - |\alpha_p^{(t)} - \beta_p^{(t)}| \precsim \lambda\|\boldsymbol{\alpha}^{(t)} - \boldsymbol{\beta}^{(t)}\|_\infty + k\varepsilon.$$

Combining all, with probability at least $1 - \delta$,

$$\|\boldsymbol{\alpha}^{(t+1)} - \boldsymbol{\beta}^{(t+1)}\|_\infty \precsim \lambda\|\boldsymbol{\alpha}^{(t)} - \boldsymbol{\beta}^{(t)}\|_\infty + k\varepsilon.$$

Based on the above recurrence relation, with probability at least $1 - \delta$,

$$\|\boldsymbol{\alpha}^{(T)} - \boldsymbol{\beta}^{(T)}\|_\infty \precsim \lambda\|\boldsymbol{\alpha}^{(T-1)} - \boldsymbol{\beta}^{(T-1)}\|_\infty + k\varepsilon \precsim \cdots \precsim \lambda^T\|\boldsymbol{\alpha}^{(0)} - \boldsymbol{\beta}^{(0)}\|_\infty + \lambda^{T-1}k\varepsilon = \lambda^{T-1}k\varepsilon,$$

which satisfies the condition of Definition 3.1.

$\square$

## C.5. Proof of Theorem 4.5

**Lemma C.4.** *Assume that* $\widetilde{\mathbf{M}} \in \mathbb{R}^{m \times m}$ *is computed by core kernel, i.e.,* $\widetilde{M}_{pq} = \mathrm{tr}\left(\frac{1}{ns}\widetilde{\mathbf{K}}_p\widetilde{\mathbf{K}}_q\right)$. *If* $s \geq c\log(n/\delta)/\varepsilon^2$, *with probability at least* $1 - \delta$,

$$-\varepsilon m\mathbf{I}_m \preccurlyeq \widetilde{\mathbf{M}} - \mathbf{M} \preccurlyeq \varepsilon m\mathbf{I}_m.$$

*Proof.* For any two indexed $p, q \in [m]$, let $\mathbf{K}_p = \boldsymbol{\Phi}_n^\top\boldsymbol{\Phi}_n$ and $\mathbf{K}_q = \boldsymbol{\Psi}_n^\top\boldsymbol{\Psi}_n$, where $\boldsymbol{\Phi}_n = [\phi_p(x_1), \cdots, \phi_p(x_n)]$ and $\boldsymbol{\Psi}_n = [\phi_q(x_1), \cdots, \phi_q(x_n)]$. Let $\boldsymbol{\Phi}_s = [\phi_p(a_1), \cdots, \phi_p(a_s)]$ and $\boldsymbol{\Psi}_s = [\phi_q(a_1), \cdots, \phi_q(a_s)]$. Consequently,

$\widetilde{\mathbf{K}}_p = (\mathbf{\Phi}_s^\top \mathbf{\Phi}_n \mathbf{\Phi}_n^\top \mathbf{\Phi}_s)^{1/2}$ and $\widetilde{\mathbf{K}}_q = (\mathbf{\Psi}_s^\top \mathbf{\Psi}_n \mathbf{\Psi}_n^\top \mathbf{\Psi}_s)^{1/2}$. Then, we have

$$
\begin{aligned}
\widetilde{M}_{pq} &= \mathrm{tr}\left( \frac{1}{ns}(\mathbf{\Phi}_s^\top \mathbf{\Phi}_n \mathbf{\Phi}_n^\top \mathbf{\Phi}_s)^{1/2}(\mathbf{\Psi}_s^\top \mathbf{\Psi}_n \mathbf{\Psi}_n^\top \mathbf{\Psi}_s)^{1/2} \right) \\
&= \mathrm{tr}\left( \left(\frac{1}{\sqrt{ns}}\mathbf{\Phi}_s^\top \mathbf{\Phi}_n\right)^{1/2} \left(\frac{1}{\sqrt{ns}}\mathbf{\Phi}_n^\top \mathbf{\Phi}_s\right)^{1/2} \left(\frac{1}{\sqrt{ns}}\mathbf{\Psi}_s^\top \mathbf{\Psi}_n\right)^{1/2} \left(\frac{1}{\sqrt{ns}}\mathbf{\Psi}_n^\top \mathbf{\Psi}_s\right)^{1/2} \right).
\end{aligned}
\tag{33}
$$

Thus, we can obtain

$$
\widetilde{M}_{pq} = \left\| \left(\frac{1}{\sqrt{ns}}\mathbf{\Phi}_n^\top \mathbf{\Phi}_s\right)^{1/2} \left(\frac{1}{\sqrt{ns}}\mathbf{\Psi}_s^\top \mathbf{\Psi}_n\right)^{1/2} \right\|_{\mathrm{F}}^2.
$$

Let the SVD of $\frac{1}{\sqrt{ns}}\mathbf{\Phi}_n^\top \mathbf{\Phi}_s$ be $\widetilde{\mathbf{U}}_1 \widetilde{\mathbf{\Lambda}}_1 \widetilde{\mathbf{V}}_1^\top$, where $\widetilde{\mathbf{U}}_1 \in \mathbb{R}^{n \times n}$, $\widetilde{\mathbf{V}}_1 \in \mathbb{R}^{s \times s}$, and $\widetilde{\mathbf{\Lambda}}_1 \in \mathbb{R}^{n \times s}$ in which the diagonal elements in the first $s \times s$ block are $\tilde{\mu}_1, \cdots, \tilde{\mu}_s$, i.e., the singular values of $\frac{1}{\sqrt{ns}}\mathbf{\Phi}_n^\top \mathbf{\Phi}_s$. Similarly, let the SVD of $\left(\frac{1}{\sqrt{ns}}\mathbf{\Psi}_s^\top \mathbf{\Psi}_n\right)^{1/2}$ be $\widetilde{\mathbf{U}}_2 \widetilde{\mathbf{\Lambda}}_2 \widetilde{\mathbf{V}}_2^\top$, and $\widetilde{\mathbf{\Lambda}}_2$ contains the corresponding singular values $\tilde{\lambda}_1, \cdots, \tilde{\lambda}_s$. Because the Frobenius norm is unitarily invariant, we have

$$
\widetilde{M}_{pq} = \left\| \widetilde{\mathbf{U}}_1 \widetilde{\mathbf{\Lambda}}_1^{1/2} \widetilde{\mathbf{V}}_1^\top \widetilde{\mathbf{U}}_2 \widetilde{\mathbf{\Lambda}}_2^{1/2} \widetilde{\mathbf{V}}_2^\top \right\|_{\mathrm{F}}^2 = \left\| \widetilde{\mathbf{\Lambda}}_1^{1/2} \widetilde{\mathbf{V}}_1^\top \widetilde{\mathbf{U}}_2 \widetilde{\mathbf{\Lambda}}_2^{1/2} \right\|_{\mathrm{F}}^2 = \left\| \widetilde{\mathbf{V}}_1^\top \widetilde{\mathbf{U}}_2 \widetilde{\mathbf{\Lambda}}_2^{1/2} \widetilde{\mathbf{\Lambda}}_1^{1/2} \right\|_{\mathrm{F}}^2 = \left\| \widetilde{\mathbf{\Lambda}}_2^{1/2} \widetilde{\mathbf{\Lambda}}_1^{1/2} \right\|_{\mathrm{F}}^2 = \sum_{i=1}^{s} \tilde{\mu}_i \tilde{\lambda}_i.
$$

Denote that the eigenvalues of $\frac{1}{n}\mathbf{\Phi}_n^\top \mathbf{\Phi}_n$ and $\frac{1}{n}\mathbf{\Psi}_n^\top \mathbf{\Psi}_n$ are $\mu_1, \cdots, \mu_n$ and $\lambda_1, \cdots, \lambda_n$, respectively. With a similar derivation, we have

$$
M_{pq} = \sum_{i=1}^{n} \mu_i \lambda_i.
$$

Letting $\{\tilde{\mu}_i\}_{i \geq s+1}$ and $\{\tilde{\lambda}_i\}_{i \geq s+1}$ be 0, by Theorem 4.1, we have

$$
|\tilde{M}_{pq} - M_{pq}| \leq \left| \sum_{i=1}^{n} (\tilde{\mu}_i \tilde{\lambda}_i - \mu_i \lambda_i) \right| \leq \left| \sum_{i=1}^{n} \tilde{\mu}_i (\tilde{\lambda}_i - \lambda_i) \right| + \left| \sum_{i=1}^{n} (\tilde{\mu}_i - \mu_i)\lambda_i \right| \leq \varepsilon \left( \sum_{i=1}^{n} \tilde{\mu}_i + \sum_{i=1}^{n} \lambda_i \right) \precsim \varepsilon.
$$

Thus, for unit vector $\boldsymbol{u} \in \mathbb{R}^m$,

$$
\|\widetilde{\mathbf{M}} - \mathbf{M}\| = \sup_{\boldsymbol{u}} |\boldsymbol{u}^\top (\widetilde{\mathbf{M}} - \mathbf{M})\boldsymbol{u}| \leq \sum_{p=1}^{m} \sum_{q=1}^{m} |u_p u_q (\tilde{M}_{pq} - M_{pq})| \leq \varepsilon \left( \sum_{p=1}^{m} |u_p| \right)^2 \leq \varepsilon m.
$$

The desirable result follows.

$\square$

**Lemma C.5** (Theorem 4.1, (Wedin, 1973))**.** *For any two $m \times m$ real matrices $\mathbf{A}$, $\mathbf{B}$, if $\mathrm{rank}(\mathbf{A}) = \mathrm{rank}(\mathbf{B}) = m$, then*

$$
\|\mathbf{B}^+ - \mathbf{A}^+\| \leq \|\mathbf{B}^+\|\|\mathbf{A}^+\|\|\mathbf{B} - \mathbf{A}\|.
$$

*Proof of Theorem 4.5.* For any unit vector $\mathbf{u} \in \mathbb{R}^m$, we have

$$
\mathbf{u}^\top \mathbf{M} \mathbf{u} = \sum_{p=1}^{m} \sum_{q=1}^{m} u_p u_q \mathrm{tr}\left( \frac{1}{n^2}\mathbf{K}_p \mathbf{K}_q \right) = \mathrm{tr}\left( \frac{1}{n}\sum_{p=1}^{m} u_p \mathbf{K}_p \right)^2 \leq \left( \mathrm{tr}\left( \frac{1}{n}\sum_{p=1}^{m} u_p \mathbf{K}_p \right) \right)^2 \precsim m.
\tag{34}
$$

Similarly, we also have $\mathbf{u}^\top \widetilde{\mathbf{M}} \mathbf{u} \precsim m$.

We use the same notation to represent the kernel weight changes at each iteration step. Then, by the optimization of MKKM-MR, we have

$$\|\boldsymbol{\alpha}^{(t+1)} - \boldsymbol{\beta}^{(t+1)}\|_\infty = \left\| \frac{(\lambda\mathbf{M} + \mathbf{D}^{(t)})^{-1}\mathbf{1}_m}{\mathbf{1}_m^\top(\lambda\mathbf{M} + \mathbf{D}^{(t)})^{-1}\mathbf{1}_m} - \frac{(\lambda\widetilde{\mathbf{M}} + \widetilde{\mathbf{D}}^{(t)})^{-1}\mathbf{1}_m}{\mathbf{1}_m^\top(\lambda\widetilde{\mathbf{M}} + \widetilde{\mathbf{D}}^{(t)})^{-1}\mathbf{1}_m} \right\|_\infty$$

$$\leq \frac{\|(\lambda\mathbf{M} + \mathbf{D}^{(t)})^{-1}\mathbf{1}_m - (\lambda\widetilde{\mathbf{M}} + \widetilde{\mathbf{D}}^{(t)})^{-1}\mathbf{1}_m\|_\infty}{\min\{\mathbf{1}_m^\top(\lambda\mathbf{M} + \mathbf{D}^{(t)})^{-1}\mathbf{1}_m, \mathbf{1}_m^\top(\lambda\widetilde{\mathbf{M}} + \widetilde{\mathbf{D}}^{(t)})^{-1}\mathbf{1}_m\}} \tag{35}$$

$$= \frac{1}{m} \cdot \frac{\|(\lambda\mathbf{M} + \mathbf{D}^{(t)})^{-1}\mathbf{1}_m - (\lambda\widetilde{\mathbf{M}} + \widetilde{\mathbf{D}}^{(t)})^{-1}\mathbf{1}_m\|_\infty}{\min\{\frac{1}{\sqrt{m}}\mathbf{1}_m^\top(\lambda\mathbf{M} + \mathbf{D}^{(t)})^{-1}\frac{1}{\sqrt{m}}\mathbf{1}_m, \frac{1}{\sqrt{m}}\mathbf{1}_m^\top(\lambda\widetilde{\mathbf{M}} + \widetilde{\mathbf{D}}^{(t)})^{-1}\frac{1}{\sqrt{m}}\mathbf{1}_m\}}$$

Because $\|\mathbf{D}^{(t)}\| \leq 1$, we have $\frac{1}{\sqrt{m}}\mathbf{1}_m^\top(\lambda\mathbf{M} + \mathbf{D}^{(t)})^{-1}\frac{1}{\sqrt{m}}\mathbf{1}_m \succsim (\lambda m + 1)^{-1} \succsim (\lambda m)^{-1}$. Moreover, $\frac{1}{\sqrt{m}}\mathbf{1}_m^\top(\lambda\widetilde{\mathbf{M}} + \widetilde{\mathbf{D}}^{(t)})^{-1}\frac{1}{\sqrt{m}}\mathbf{1}_m \succsim (\lambda m)^{-1}$. Combining Eq.(35), we have

$$\|\boldsymbol{\alpha}^{(t+1)} - \boldsymbol{\beta}^{(t+1)}\|_\infty \precsim \lambda\|(\lambda\mathbf{M} + \mathbf{D}^{(t)})^{-1}\mathbf{1}_m - (\lambda\widetilde{\mathbf{M}} + \widetilde{\mathbf{D}}^{(t)})^{-1}\mathbf{1}_m\|_\infty$$

$$\leq \lambda\|(\lambda\mathbf{M} + \mathbf{D}^{(t)})^{-1}\mathbf{1}_m - (\lambda\widetilde{\mathbf{M}} + \widetilde{\mathbf{D}}^{(t)})^{-1}\mathbf{1}_m\|$$

$$\leq \sqrt{m}\lambda\|(\lambda\mathbf{M} + \mathbf{D}^{(t)})^{-1} - (\lambda\widetilde{\mathbf{M}} + \widetilde{\mathbf{D}}^{(t)})^{-1}\|$$

$$\leq \sqrt{m}\lambda\|(\lambda\mathbf{M} + \mathbf{D}^{(t)})^{-1}\|\|(\lambda\widetilde{\mathbf{M}} + \widetilde{\mathbf{D}}^{(t)})^{-1}\|\|\lambda\mathbf{M} + \mathbf{D}^{(t)} - (\lambda\widetilde{\mathbf{M}} + \widetilde{\mathbf{D}}^{(t)})\|$$

(By Lemma C.5.)

$$\precsim \frac{\sqrt{m}}{\lambda}(\lambda\|\mathbf{M} - \widetilde{\mathbf{M}}\| + \|\mathbf{D}^{(t)} - \widetilde{\mathbf{D}}^{(t)}\|)$$

(By the assumption that $\mathbf{M}, \widetilde{\mathbf{M}}$ have full ranks.)

$$\precsim \frac{\sqrt{m}}{\lambda}\left(\lambda\varepsilon m + \max_{q\in[m]}\left|\mathrm{tr}(\frac{1}{n}\mathbf{K}_q) - \mathrm{tr}(\frac{1}{\sqrt{ns}}\widetilde{\mathbf{K}}_q)\right| + \max_{q\in[m]}\left|\mathcal{T}_n(K_q, \{\hat{h}_j^{\boldsymbol{\alpha}^{(t)}}\}_{j=1}^k) - \mathcal{T}_s(\widetilde{K}_q, \{\tilde{h}_j^{\boldsymbol{\beta}^{(t)}}\}_{j=1}^k)\right|\right)$$

$$\precsim \frac{\sqrt{m}}{\lambda}(\lambda\varepsilon m + k\varepsilon + \|\boldsymbol{\alpha}^{(t)} - \boldsymbol{\beta}^{(t)}\|_\infty)$$

$$\precsim \|\boldsymbol{\alpha}^{(t)} - \boldsymbol{\beta}^{(t)}\|_\infty + \varepsilon. \tag{36}$$

We can obtain the desirable result based on the above recurrence relation. The proof is complete.

$\square$

# D. More Experimental Results

## D.1. Approximation Effect of Core Kernel on Kernel Weights

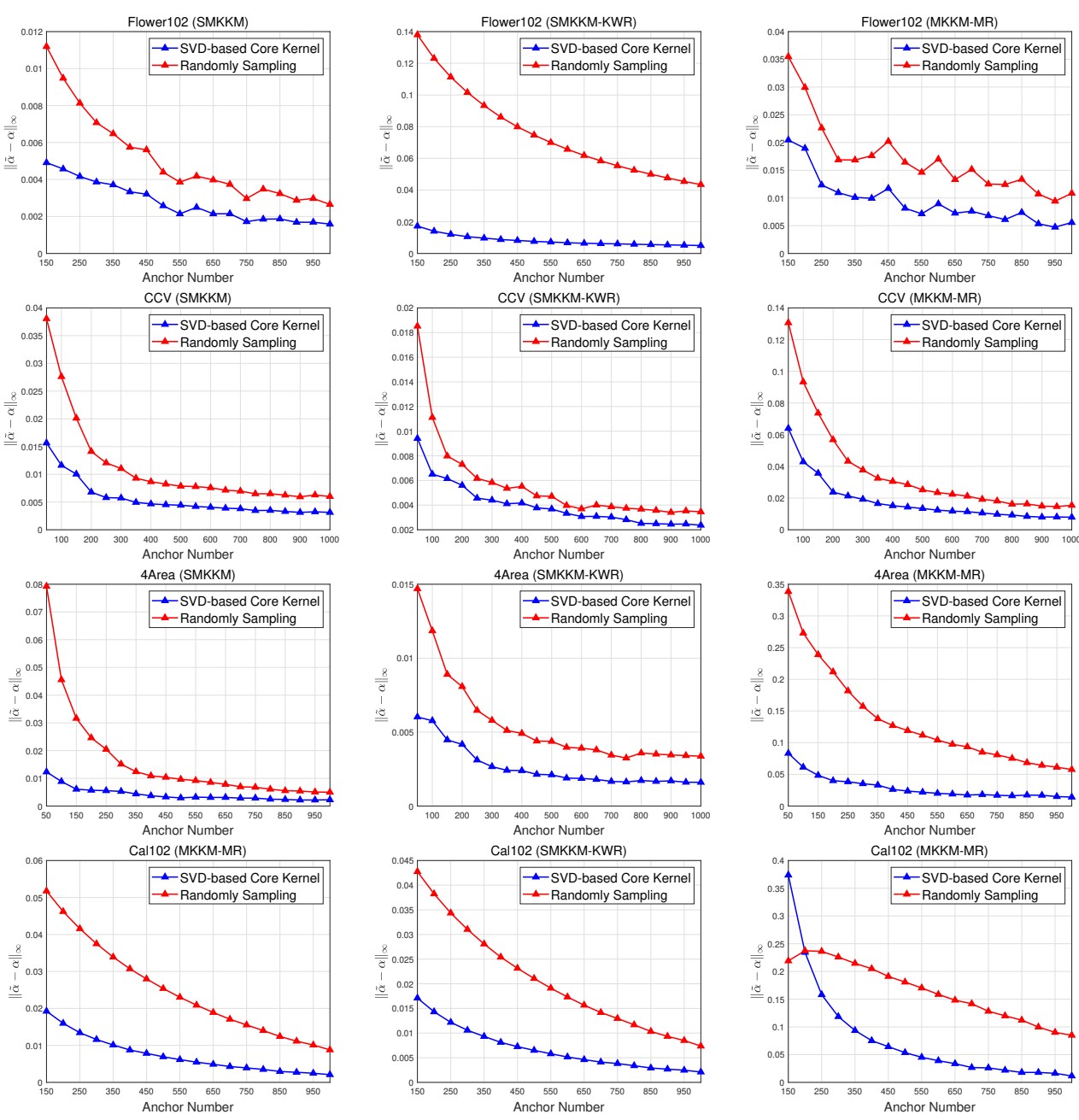

*Figure 3.* The proposed SVD-CK is illustrated through a diagram showing the kernel weight approximation performance. The blue curve represents the kernel weight approximation error constructed using SVD-CK. It can be observed that as $s$ increases, the approximation error decreases rapidly, enabling the weights obtained by the three MKC methods on SVD-CK to closely approximate those on the original kernel matrices. For comparison, the red curve represents the kernel weight approximation error based on random sampling of the kernel matrix. SVD-CK demonstrates a clear advantage in kernel weight approximation.

## D.2. Information of Kernel Datasets

The detailed information of six large-scale datasets is listed in Table 3, and their URL links are as

- Flower17: http://www.robots.ox.ac.uk/~vgg/data/flowers/17/

- Digit: http://ss.sysu.edu.cn/py/

- CCV: http://www.ee.columbia.edu/ln/dvmm/CCV/

- Flower102: http://www.robots.ox.ac.uk/~vgg/data/flowers/102/

- 4Area: (Perozzi et al., 2014)

- Cal102: http://www.vision.caltech.edu/ImageDatasets/Caltech101/

*Table 3.* Six small-scale kernel datasets.

| Dataset | Number of | | |
| | Samples | Kernels | Clusters |
| --- | --- | --- | --- |
| Flower17 | 1360 | 7 | 17 |
| DIGIT | 2000 | 3 | 10 |
| CCV | 6773 | 3 | 20 |
| Flower102 | 8189 | 4 | 102 |
| 4Area | 4236 | 2 | 4 |
| Cal102 | 1530 | 25 | 102 |

## D.3. Information of Comparison Methods

Detailed information of comparison methods is as follows.

- **1) Robust Multi-View $k$-Means Clustering (RMKMC)**(Cai et al., 2013): RMKMC is a robust large-scale multi-view $k$-means clustering algorithm.

- **2) Large-Scale Multi-View Subspace Clustering (LMVSC)**(Kang et al., 2020): LMVSC constructs a similarity matrix using selected anchor points to reduce redundant computations in subspace clustering.

- **3) One-Pass Multi-View Clustering (OPMC)**(Liu et al., 2021): OPMC eliminates the non-negative constraints in non-negative matrix factorization and integrates all views to achieve a unified partition.

- **4) Auto-Weighted Multi-View Clustering (AWMVC)**(Wan et al., 2024): AWMVC derives coefficient matrices from the base matrices of different dimensions and fuses them to obtain the optimal consensus matrix.

## D.4. Whole Experimental Results on Large-Scale Datasets

*Table 4.* Results of large-scale experiments

| Datasets | CIFAR10 | MNIST | Winnipeg |
|---|---|---|---|
| ACC (%) | | | |
| RMKMC | 82.95 | 85.60 | 62.25 |
| LMVSC | 49.50 | 86.14 | 60.25 |
| OPMC | 69.59 | 84.92 | 53.53 |
| AWMVC | 80.90 | 83.23 | 53.66 |
| SMKKM (CK) | 97.46 | 99.02 | 62.09 |
| SMKKM-KWR (CK) | 98.15 | 99.01 | **62.10** |
| MKKM-MR (CK) | **99.28** | **99.15** | 59.24 |
| NMI (%) | | | |
| RMKMC | 82.07 | 81.05 | 49.43 |
| LMVSC | 45.04 | 84.75 | 51.94 |
| OPMC | 83.81 | 82.67 | 50.82 |
| AWMVC | 76.38 | 80.76 | 38.86 |
| SMKKM (CK) | 97.53 | 97.00 | 54.14 |
| SMKKM-KWR (CK) | 97.78 | 96.96 | 54.12 |
| MKKM-MR (CK) | **98.07** | **97.33** | **59.24** |
| Purity (%) | | | |
| RMKMC | 86.78 | 86.74 | 65.98 |
| LMVSC | 58.96 | 89.14 | 70.31 |
| OPMC | 87.82 | 85.45 | 64.72 |
| AWMVC | 84.00 | 87.41 | 67.74 |
| SMKKM (CK) | 97.96 | 99.02 | 79.24 |
| SMKKM-KWR (CK) | 98.43 | 99.01 | **79.71** |
| MKKM-MR (CK) | **99.28** | **99.15** | 69.25 |
| Time (s) | | | |
| RMKMC | 162.09 | 155.16 | 297.40 |
| LMVSC | 16.22 | 67.44 | 142.63 |
| OPMC | 27.56 | 49.94 | 20.29 |
| AWMVC | 203.01 | 64.78 | 59.77 |
| SMKKM (CK) | 47.84 | 65.18 | 288.06 |
| SMKKM-KWR (CK) | 43.61 | 65.77 | 248.51 |
| MKKM-MR (CK) | 38.99 | 62.26 | 259.24 |

