# OpenReview forum: "COKE: Core Kernel for More Efficient Approximation of Kernel Weights in Multiple Kernel Clustering"
_ICML.cc/2025/Conference — ICML 2025 poster_

### Official Review · Reviewer_s6E6 · 2025-03-07

**Overall Recommendation:** 4

**Summary:**

The paper introduces a novel approach to multiple kernel clustering by proposing the concept of core kernel. This method aims to reduce the computational complexity of clustering large datasets while preserving performance.

**Main Contributions:**

1.	Core Kernel Definition: The core kernel is defined as a set of smaller-sized kernel matrices that can approximate the original kernel weights with a (1+$\varepsilon$)-approximation guarantee.

2.	Algorithmic Contribution: The paper propose an algorithm based on SVD to construct core kernel efficiently. This algorithm significantly reduces the size of the kernel matrices while preserving essential information for clustering.

3.	Theoretical Guarantees: The paper provides theoretical proofs (Theorem 4.4 and Theorem 4.5) that establish the effectiveness of the core kernels in approximating the original kernel weights under certain assumptions, such as the existence of a spectral gap.

**Claims And Evidence:**

The overall logic of the paper is quite clear.

**Introduction**

It is important to emphasize the differences between this paper and the related work, highlighting the innovative aspects of the proposed method.

**Motivation Needs More Detailed Elaboration**

The motivation of the paper requires more detailed elaboration to clearly illustrate the significance of kernel weights. Specifically, it should provide a more thorough explanation of why kernel weights are crucial and how they impact the final clustering results.

**Essential References Not Discussed:**

As far as I know, this paper has basically discussed all the essential related work.

**Experimental Designs Or Analyses:**

The experimental design is relatively reasonable; however, there are several issues:

1. Why does the chosen value of the number of anchor points s not correspond to the conclusion of the number s obtained in the theorem?

2. It is recommended to include a comparison of clustering performance between the kernel weights \alpha obtained from SMKKM and MKKM, as well as the kernel weights \tilde{ \alpha } proposed in the paper.

**Methods And Evaluation Criteria:**

The methods section is well-organized, and the explanation regarding the core kernel is very clear. However, concerning matrix T, the paper mentions using uniform sampling, and it may be worth considering leverage score sampling as well.

**Other Comments Or Suggestions:**

It is recommended that the differences and advantages of the proposed method be elaborated in detail compared to the methods in “Consistency of Multiple Kernel Clustering”.

**Other Strengths And Weaknesses:**

Strength: the paper provides rigorous mathematical proofs to support their claims, ensuring that the proposed SVD-CK method has a strong theoretical basis.

Weakness : the paper lacks performance experiments for the proposed method on small datasets.

**Questions For Authors:**

The paper uses the same anchor point set for all kernel approximations. Has there been any consideration of approximating different kernels with different anchor points?

In this paper, the number and quality of anchor points are relatively important. The theorems presented only provide conclusions regarding the number of anchor points. Has there been any consideration of whether the quality of anchor points can be theoretically evaluated?

**Relation To Broader Scientific Literature:**

The key contributions of this paper build upon prior research in clustering and kernel methods, particularly enhancing the understanding of anchor point selection and its impact on clustering performance.

**Theoretical Claims:**

The theoretical theorems and proofs presented in the paper are clear and rigorous. It is recommended to discuss the relationship between the clustering subspace and the weights in the remark section of Theorem 3.3, emphasizing the motivation behind it.

---

> ### Author Rebuttal · Authors · 2025-04-01
>
> We appreciate the comments of Reviewer s6E6 and have responded to them individually.
>
> ### Q1: Why are kernel weights crucial?
>
> A2: In many previous studies, it can be observed that the kernel weights of multiple kernel clustering (MKC) algorithms have a significant impact on the learning performance of the algorithms. For example, in a classical reference [1] (https://ojs.aaai.org/index.php/AAAI/article/download/10249/10108), the authors compare various clustering metrics of the optimal single kernel, average kernel, and various MKC algorithms, showing that their clustering performances differ greatly. Different optimization principles for kernel weights, tailored to different datasets and learning scenarios, result in significant differences in clustering performance. According to the no free lunch theorem, for a specific problem, we need to adjust the algorithm correspondingly. The same applies to MKC algorithms, where it is essential to develop targeted kernel weight design schemes. The above empirical evidence fully demonstrates that the kernel weights of base kernels have a crucial impact on improving the performance of MKC algorithms.
>
> ### Q2:  Discuss the relationship between the clustering subspace and the weights in the remark of Theorem 3.3.
>
> A2: For the original MKC algorithm, $\mathbf{H}$ is the sample embedding used to obtain the final clustering results, while the approximate embedding proposed in this paper is $\widetilde{\mathbf{H}}$. We aim to demonstrate through Theorem 3.3 that the proposed method and the original algorithm can achieve similar clustering performance. To measure the difference between $\mathbf{H}$ and $\widetilde{\mathbf{H}}$, the best way is to compare the distance between their projections. We will include a more detailed description in the revised version.
>
> ### Q3: Why does the chosen value of the number of anchor points $s$ not correspond to the conclusion obtained in the theorem?
>
> A3: The main reasons are as follows: Firstly, the theoretical derivation is based on some general assumptions, and the conclusions obtained are valid under all conditions. However, in empirical observations, factors such as the distribution of the dataset, the kernel function, and other elements affecting the approximation effect may result in a smaller required value of $s$ in practice. Secondly, the boundary of the theoretical derivation is constrained by the matrix concentration inequality shown in Theorem B.3 of the appendix. In real situations, the concentration of the dataset used in experiments may be smaller than the upper bound presented in Theorem B.3.
>
> ### Q4: Compare the clustering performance between the proposed method and the original algorithm.
>
> A4: We compared the NMI of the proposed algorithm and the original algorithm on SMKKM and MKKM-MR using two datasets, Flower17 and DIGIT. The number of anchors ranged from 100 to 1000, with a step size of 100. The final experimental results are shown in the figure (https://anonymous.4open.science/r/ICML2025-8D7F/NMI_RES.svg). It can be observed that when the number of anchors is around 500, the proposed algorithm achieves similar clustering performance to the original.
>
> ### Q5: Elaborate on the differences between the proposed method and the algorithm in [2].
>
> A5: The method presented in this paper has the following advantages: 1. Wider Applicability: This paper proves that the proposed method can be widely applied to the kernel weight approximation of three types of MKC algorithms, whereas the method in [2] can only approximate one type of MKC algorithm. 2. Better Approximation Effect: As shown in Figures 2 and 3, the approximation effect of the proposed method is better than that of the method in [2].
>
> [2] Consistency of Multiple Kernel Clustering. ICML 2023.
>
> ### Q6: Approximate different kernels with different anchor points.
>
> A6: We have attempted to use different anchor points for different base kernels to perform approximation. However, due to the inconsistency of the anchor sets, the approximation effect turned out to be rather poor. In multi-view clustering, some studies have employed different anchor points for different views and achieved promising clustering results. However, these works lack theoretical guarantees for approximation, and they do not provide theoretical insights that are applicable to our results. The reviewer’s suggestion is highly insightful for us, and we will attempt to address this issue in our future work.
>
> ### Q7: Evaluate the quality of anchor points theoretically.
>
> A7: Compared with uniform sampling, importance sampling methods (such as leverage scores) can select more representative anchor points, allowing good approximation results with fewer anchors. However, this leads to inconsistency in the anchor points of different base kernels, which affects the final kernel weight approximation. This situation is similar to that described in Question 6, and we will investigate this issue in future work.

---

### Official Review · Reviewer_hzsf · 2025-03-10

**Overall Recommendation:** 4

**Summary:**

This paper introduces a novel concept called the "core kernel," which aims to approximate kernel weights in multiple kernel clustering (MKC) algorithms by running them on smaller-scale base kernel matrices. The core kernel, with a size of $\widetilde{\mathcal{O}}(1/\varepsilon)$, achieves a $1+\varepsilon$-approximation. The authors propose a core kernel construction method based on singular value decomposition (SVD-CK) and prove its applicability to three mainstream MKC algorithms (SMKKM, SMKKM-KWR, and MKKM-MR). Theoretical analysis shows that using the core kernel reduces the time complexity of MKC algorithms from $\mathcal{O}(n^3)$ to $\mathcal{O}(s^3)$. Experiments on benchmark datasets validate the approximation performance of SVD-CK on kernel weights, while tests on large-scale datasets demonstrate the efficiency of the scalable extension. The core contribution lies in providing a theoretically guaranteed framework for efficient large-scale extensions of MKC algorithms.

**Claims And Evidence:**

Key claims are supported by experiments and theory:
Claim 1: SVD-CK can effectively approximate kernel weights.
Evidence: Theorems 4.4 and 4.5 demonstrate that the SVD-CK method can produce a $1+\varepsilon$-approximation core kernel set for SMKKM, SMKKM-KWR, and MKKM-MK. Figures 2 show that the kernel weight error of SVD-CK decreases rapidly as the anchor number $s$ increases, significantly outperforming random sampling (blue vs. red curves).

Claim 2: The core kernel enables large-scale extensions for MKC.
Evidence: Algorithm 1 first obtains approximate weights through the SVD-CK method, then combines the base similarity matrices using these weights, and finally derives the cluster indicator matrix via SVD on the combined matrix. Theorem 3.3 proves that if the algorithm inputs are identical, the cluster indicator matrix obtained through Algorithm 1 can arbitrarily approximate the cluster indicator matrices obtained by other MKC algorithms. Table 2 shows that core kernel-based MKC algorithms (e.g., SMKKM) achieve high clustering performance (NMI >97%) on datasets with 50k samples, with reasonable time costs.

**Essential References Not Discussed:**

The paper essentially covers the important relevant literature.

**Experimental Designs Or Analyses:**

Reproducibility: Hyperparameter settings (e.g., $\sigma^2$ for Gaussian kernels) are clear, but code or implementation details for anchor sampling are missing.
Statistical Analysis: Experiments are repeated 30 times for averaging, reducing randomness, but variance or significance tests are not reported.

**Methods And Evaluation Criteria:**

Method Rationality: SVD-CK compresses kernel matrices via anchor sampling and SVD, aligning with coreset principles.
Evaluation Criteria: Experiments cover benchmark datasets (e.g., Flower17, CCV) and large-scale datasets (e.g., Winnipeg), with metrics including NMI, ACC, and runtime.

**Other Comments Or Suggestions:**

No.

**Other Strengths And Weaknesses:**

Strengths:
1. Novelty: Extends coreset ideas to kernel weight approximation with solid theoretical analysis.
2. Practicality: The method is simple and scales to datasets with over 300k samples.
Weaknesses:
1. Experiments only use the Gaussian kernel function, lacking validation on other kernel functions.
2. Time costs for core kernel construction (e.g., SVD complexity) are not discussed.

**Questions For Authors:**

Q1: Assumption 3.2 assumes eigenvalue gaps, but real-world data may violate this. If eigenvalue gaps approach zero, does Theorem 3.3 still hold? Please discuss this scenario.

Q2: Is the construction time of SVD-CK significantly higher than random sampling? Please include runtime comparisons.

**Relation To Broader Scientific Literature:**

The paper connects to:
1. Coresets in clustering (Har-Peled & Mazumdar, 2004).
2.MKC algorithms (Huang et al., 2012; Liu, 2022).
3.Large-scale kernel methods (Wang et al., 2019).

**Theoretical Claims:**

Theorem 3.3
Statement: Under Assumption 3.2 (eigenvalue gap), the subspace difference between the clustering indicator matrix $\widetilde{\mathbf{H}}$ from the core kernel (via Algorithm 1) and the original $\mathbf{H}$ satisfies.
Potential Issues: Validity of Assumption 3.2: Eigenvalue gaps may not hold for high-dimensional/noisy data where eigenvalues are densely distributed.

Theorem 4.1
Statement: The singular values of the uniformly sampled matrix approximate the eigenvalues of the original kernel matrix with $\varepsilon$-error, high probability.
Potential Issues: Uniform Sampling Assumption: May fail for non-uniform data distributions.

---

> ### Author Rebuttal · Authors · 2025-04-01
>
> We sincerely appreciate the work of Reviewer hzsf and respond to the review comments as follows.
>
> ### Q1: Assumption 3.2 assumes eigenvalue gaps, but real-world data may violate this. If eigenvalue gaps approach zero, does Theorem 3.3 still hold? Please discuss this scenario.
>
> A1: The reason for making this assumption is that, in the perturbation analysis of matrix eigenvectors, the eigenvalues of a matrix are often considered to be isolated, and thus the eigenvalue gap is always greater than 0. Since the proof of Theorem 3.3 involves the results from matrix perturbation theory (Lemma C.1), we have to make this assumption in this paper.
>
> In practical situations, even if there exist cases where the eigenvalue gap is 0, we conjecture that the above conclusions from matrix perturbation theory still hold. However, this would require more powerful mathematical tools. Currently, it is necessary to strengthen the conclusions regarding eigenvector perturbations in matrix perturbation theory to avoid this assumption.
>
>
> ### Q2: Is the construction time of SVD-CK significantly higher than random uniform sampling? Please include runtime comparisons.
>
> A2: The method of constructing the base kernel matrix using random uniform sampling requires a time complexity of $\mathcal{O}(s^2)$ since it only needs to sample the indices of anchor points. In contrast, the construction method of SVD-CK provided by Algorithm 2 requires a time complexity of $\mathcal{O}(ns^2)$. However, this process only needs to be executed once, so it does not significantly increase the overall time of the algorithm. We compared the time taken to obtain the final kernel weights using random sampling and SVD-CK, as shown in the figure (https://anonymous.4open.science/r/ICML2025-8D7F/time_cost.svg). It can be seen that the total time for obtaining kernel weights using the two construction methods is roughly the same. This fully demonstrates that the proposed SVD-CK in this paper is not significantly less efficient than random uniform sampling.

---

### Official Review · Reviewer_qRUV · 2025-03-10

**Overall Recommendation:** 4

**Summary:**

This paper first proposes the concept of core kernel, and proposes a core kernel construction method based on singular value decomposition, and proves that it meets the core kernel definition of three mainstream MKC algorithms. The correctness of the theoretical results and the effectiveness of the proposed method are verified on multiple benchmark datasets.

## update after rebuttal

I keep my positive score.

**Claims And Evidence:**

yes.

**Essential References Not Discussed:**

no

**Experimental Designs Or Analyses:**

no.

**Methods And Evaluation Criteria:**

yes.

**Other Comments Or Suggestions:**

please see questions.

**Other Strengths And Weaknesses:**

The definition of the core kernel is interesting.

The paper provides solid theoretical guarantees.

**Questions For Authors:**

1. What is the relationship between Algorithm 1($k$-means after using SVD) and spectral clustering?

2. How is the proposed method  (algorithm 2) related to Neystrom?

3. What is the intuition of Algorithm 2 (such as what is the meaning of Kp~)?

4. Are the eigenvalues ​​of Assumption 3.2 sorted?

5. Theorem 3.3 and 4.1 require a large $s$. (Flower17 in Figure 1 only has 1000 points?)

6. the approximation effect of $\frac{1}{\sqrt(ns)}P$ is much better than $\frac{1}{s}W$. Why is this? W is the eigenvalue calculated directly, while $\frac{1}{\sqrt(ns)}P$ is estimated using singular. Why is it better?

7. Figure 1 only shows the difference $| a-b|$, but how big is $\lambda_j(\frac{1}{n}K)$? Can you show $\frac{| a-b|}{\lambda_j(\frac{1}{n}K)}$ in a new figure?

8. What kernels were used? Are they all Gaussian kernels? How are their parameters set?

**Relation To Broader Scientific Literature:**

Inspired by the well-known core set in clustering algorithms, a definition of the core kernel is introduced to obtain kernel weights similar to those obtained using the standard basis kernel matrix.

**Theoretical Claims:**

no.

---

> ### Author Rebuttal · Authors · 2025-03-31
>
> We greatly appreciate the feedback of Reviewer qRUV and have addressed each comment point by point, as detailed below.
>
>
> ### Q1: The relationship between Algorithm 1 and spectral clustering.
>
> A1: Assume that the consensus kernel matrix obtained by the original multiple kernel clustering (MKC) is $\mathbf{K}_ {\boldsymbol{\alpha}}$. The proposed Algorithm 1 is used to approximate kernel $k$-means on $\mathbf{K}_ {\boldsymbol{\alpha}}$. Specifically, we utilize the first $k$ singular vectors of matrix $\mathbf{P}_ {\tilde{\boldsymbol{\alpha}}}$ instead of the first $k$ eigenvectors of matrix $\mathbf{K}_ {\boldsymbol{\alpha}}$ to obtain clustering results similar to those of the original MKC algorithm. Our proposed algorithm is different from spectral clustering. However, our algorithm is similar to spectral clustering in the final step, where, after obtaining the clustering indicator matrix, $k$-means clustering is used to get the final clustering results.
>
> ###  Q2: The relationship between Algorithm 2 and Nyström approximation?
>
> A2: Algorithm 2 and the Nyström method are both methods used to approximate the spectrum of kernel matrices, but they focus on different aspects as follows.
>
> Different usage: The Nyström algorithm mainly approximates the kernel $k$-means clustering results on a single kernel through the approximated eigen-decomposition of the kernel matrix. In contrast, our method is mainly used to approximate the kernel weights in MKC.
>
> Different output: The Nyström method outputs an $n\times n$ kernel matrix with low rank. However, the size of the kernel matrices outputted by our algorithm is $s\times s$. Therefore, the kernel matrix output by our algorithm is more suitable for approximating the kernel weights in the MKC algorithm.
>
> ### Q3: What is the intuition of Algorithm 2?
>
> A3: Our intuition is that it is possible to approximate the weights of MKC on the original base kernel matrix using several base kernel matrices whose sizes are independent of the number of samples $n$. The kernel matrix $\widetilde{\mathbf{K}}_p$ output by Algorithm 2 can precisely approximate the spectrum of the original kernel matrix, and its size is $s\times s$, which meets our conception.
>
> ### Q4: Are the eigenvalues of Assumption 3.2 sorted?
>
> A4: Yes, the eigenvalues described in Assumption 3.2 are sorted in descending order, which is a common assumption in kernel learning theory. To avoid unnecessary confusion, we will clarify this point in the revised version.
>
> ### Q5: Theorem 3.3 and Theorem 4.1 require a large $s$, but Flower17 in Figure 1 only has 1000 points.
>
> A5: The error bound derived theoretically may, for various reasons, be worse than the results observed empirically. However, empirical observations are always consistent with the theoretical results. The main reason for this phenomenon is that, under the necessary assumptions, the theoretically derived results hold universally under any conditions. In contrast, empirical observations fix certain conditions (such as the data distribution and the kernel function, etc.), which may lead the experiments to exhibit smaller errors. In the future, we also hope to obtain tighter error bounds under some reasonable assumptions, making them closer to the results of empirical observations.
>
> ### Q6: Why is the approximation effect of $\frac{1}{\sqrt{ns}} \mathbf{P}$ is much better than $\frac{1}{s} \mathbf{W}$?
>
> A6: The theoretical proof of why the approximation effect is better is also an issue we are currently researching. At present, we provide the following explanation for this phenomenon: $\frac{1}{\sqrt{ns}} \mathbf{P}$ is constructed from the entire training set and the sampled anchor set, whereas $\frac{1}{s} \mathbf{W}$ is constructed solely from the anchor set. It can be seen that compared to $\frac{1}{s} \mathbf{W}$, $\frac{1}{\sqrt{ns}} \mathbf{P}$ contains more information about the training set, which is why the approximation effect is better.
>
> ### Q7: Draw the graph of the relative eigenvalue approximation errors $\frac{|a-b|}{\lambda_j(\mathbf{K}/n)}$.
>
> A7: We have already plotted the graph of the relative eigenvalue approximation errors of the two methods. Please refer to the anonymous link https://anonymous.4open.science/r/ICML2025-8D7F/eigen_appro.svg. It can be seen that the relative approximation error of eigenvalues caused by SVD is much smaller than that of uniform sampling.
>
> ### Q8: What kernels were used? Are they all Gaussian kernels? How are their parameters set?
>
> A8: For small-scale datasets, we used publicly available multiple kernel datasets. Relevant researchers carefully construct these datasets and make them available for public download. As for large-scale datasets, as described in Section 5.3, we employed the Gaussian kernel function and provided the corresponding parameters based on previous research experience.

---

> > ### Comment · Reviewer_qRUV · 2025-04-02
> >
> > Thanks for your response, I will keep my score.

---

### Decision · Program_Chairs · 2025-05-01

**Decision:**

Accept (poster)

**Comment:**

All Reviewers agree on the novelty and theoretical contribution of this paper. Therefore, I recommend for Accept.